# Mortality and predictors among HIV-TB co-infected patients in Ethiopia: A systematic review and meta-analysis

Wubet Tazeb Wondie[1]*, Chalachew Adugna Wubneh[2], Bruck Tesfaye Legesse[3], Gebrehiwot Berie Mekonen[4], Alemu Birara Zemariam[5], Zenebe Abebe Gebreegziabher[6], Gezahagn Demsu Gedefaw[7], Gemechu Gelan Bekele[8], Belay Tafa Regassa[9]

1 Department of Pediatrics and Child Health Nursing, College of Health Sciences and Referral Hospital, Ambo University, Ambo, Ethiopia, 2 Department of Pediatrics and Child Health Nursing, School of Nursing, College of Medicine and Health Sciences, University of Gondar, Gondar, Ethiopia, 3 Department of Pediatrics and Neonatal Nursing, School of Nursing and Midwifery, College of Health Sciences, Wollega University, Nekemet, Ethiopia, 4 Department of Pediatrics and Child Health Nursing, College of Health Sciences, Debre Tabor University, Debre Tabor, Ethiopia, 5 Department of Pediatrics and Child Health Nursing, School of Nursing, College of Medicine and Health Sciences, Woldia University, Woldia, Ethiopia, 6 Department of Epidemiology and Biostatistics, School of Public Health, Debre Berhan University, Debre Berhan, Ethiopia, 7 Department of Neonatal Health Nursing, School of Nursing, College of Medicine and Health Sciences, University of Gondar, Gondar, Ethiopia, 8 Department of Midwifery, College of Health Sciences, Madda Walabu University, Shashamene, Ethiopia, 9 Department of Medical Laboratory Science, College of Health Sciences and Referral Hospital, Ambo University, Ambo, Ethiopia

* wubettazeb27@gmail.com

## Abstract

### Background

HIV-TB co-infection poses a significant public health threat, notably in sub-Saharan Africa including Ethiopia. Despite this public health problem, studies in Ethiopia regarding the mortality of HIV-TB co-infection patients have been inconsistent, and the overall estimate of mortality was not determined. Accordingly, this meta-analysis aims to assess the magnitude of mortality and predictors among HIV-TB co-infected patients in Ethiopia.

### Methods

A search of the literature was conducted from three databases (PubMed, Global Index Medicus, and CINHAL), and other sources (Google Scholar, Google, Worldwide Science). All observational studies that reported the mortality of HIV-TB co-infected patients in Ethiopia were included. Joanna Briggs Institute's (JBI) quality appraisal checklist was used to assess the quality of studies. Effect sizes were pooled using the random effects model. Heterogeneity was assessed using Cochrane Q and $I^2$ test statistics, and the prediction interval was determined. Subgroup analysis was conducted by region. To examine the presence of an influential study, a sensitivity analysis was done. Egger's test was used to check publication bias. A non-parametric trim and fill analysis was carried out.

### Results

A total of 886 studies were identified, using database searches and keywords. Of these, 37 met the criteria for inclusion. The pooled proportion of mortality among HIV-TB co-infected

**Data Availability Statement:** All data analyzed and used for this study are available as Supporting information.

**Funding:** The author(s) received no specific funding for this work.

**Competing interests:** The authors have declared that no competing interests exist.

**Abbreviations:** AIDS, Acquired Immunodeficiency Syndrome; ART, Anti-Retroviral Therapy; CPT, Co-trimoxazole Preventive Therapy; HIV, human immunodeficiency virus; WHO, World Health Organization.

patients was found to be 18.42% (95% CI:14.27–22.57). In the subgroup analysis, the highest mortality was observed in the Tigray region at 31.86% (95% CI: 7.69–56.03), and the lowest mortality was reported in two general studies in Ethiopia 11.95 (95% CI: 4.19–19.00). From the examined 20 predictors, only four predictors such as Anaemia (HR = 2.25, 95% CI: 1.65–3.07), Poor adherence to ART (HR = 2.42, 95% CI: 1.39–4.21), not taking co-trimoxazole preventive therapy (HR = 1.87, 95% CI: 1.28–2.73), and extrapulmonary tuberculosis (HR = 1.23, 95% CI: 1.01–1.51) were significant predictors.

## Conclusions

In Ethiopia, 18.42% of HIV-TB co-infected patients died. Anaemia, poor adherence, not taking CPT, and extrapulmonary tuberculosis were found to be significant predictors. Hence, the concerned stakeholders need to expand and strengthen the HIV-TB collaborative services and attention should be given to patients presented with the aforementioned predictors.

## Trial registration

This meta-analysis has been registered in PROSPERO with registration number CRD42023466558.

## Introduction

Human Immunodeficiency Virus (HIV) and Tuberculosis (TB) infection remain a major public health problem, notably in Sub-Sahara Africa [1]. Patients co-infected with HIV and TB are threefold more likely to die during treatment and continue to suffer from various opportunistic infections [2], which contributes to significant morbidity and mortality [3, 4]. According to the Global TB Report, an estimated 167,000 deaths from HIV-TB occurred in 2022 [5]. Additionally, the World Health Organization (WHO) reported 190, 000 deaths in 2021, with 11% of these deaths occurring in children [6]. This indicates that HIV-TB co-infection accounts for 30% of the total AIDS-related mortality [2, 7].

A global review indicated that HIV-TB co-infection resulted in 2.7 death per 100, 000 individuals, with notably high mortality rates observed in Lesotho and Zambia, at 168 & 53 death per 100,000 patients, respectively [8]. Globally the mortality of HIV-TB co-infected patients is (11%), with the highest mortality reported in Africa (14%) [9]. In Sub-Saharan Africa, 36% of HIV-TB co-infection-related deaths occurred among children [10]. In addition to the aforementioned burden, TB contributes to 24.9% of in-hospital mortality among HIV-infected patients [11], and the co-infection of these diseases imposes a significant financial burden on countries with high rates of HIV-TB [5]. In Ethiopia, the estimated costs are $78 for HIV and $115 for TB per patient [12].

Several studies investigated the risk factors for mortality in HIV-TB co-infected. As the studies showed that the factors were low CD4 count, TB Preventive Therapy, risky behaviors [8, 9, 13–15], Poor nutritional status, not taking Co-trimoxazole Preventive Therapy (CPT), male sex, being female sex worker, functional status (being bedridden and ambulatory) [7, 8], early initiation of Anti-Retroviral Therapy (ART) for TB patients [7, 16], extra-pulmonary tuberculosis, residence, poor adherence, non-disclosure, treatment failure, Multi-Drug Resistance (MDR) TB, opportunistic infection, and WHO stages (stage I, II, III) [7, 13]. To address

the dual burden of HIV and TB, WHO has established a global framework for a collaborative strategic program [17], and has recommended a test and treat all strategy since 2016. Additionally, HIV-TB control programs are intensifying their joint efforts to enhance integrated service delivery, case finding, TB preventive therapy, infection control, and the provision of ART and anti-TB medications [2, 7, 8]. Despite these measures, and improved ART coverage in Ethiopia, the rate of HIV-TB co-infection is increasing [6, 7], and the morbidity and mortality associated with this co-infection are not decreasing [7, 18, 19].

Ethiopia is one of the countries with a high burden of HIV-TB co-infection [5], and various individual studies have been conducted in different parts of the country regarding the mortality of co-infected patients. However, the mortality rate associated with HIV-TB co-infection varies across these studies, ranging from 4.4% [20] to 80.45% [21]. These findings are fragmented and do not provide comprehensive and robust conclusions regarding the mortality of co-infected patients at a national level. Current evidence-based findings are necessary to inform health program planners and policy-makers. While there is a general understanding that HIV and TB are interrelated and contribute to increased mortality from one another, to the best of our knowledge, no comprehensive study has estimated the pooled mortality rate of HIV-TB co-infected patients. Therefore, this systematic review and meta-analysis aims to deliver comprehensive findings on the mortality rates and predictors affecting co-infected patients in Ethiopia. The result of this review will provide valuable insights for decision-makers, and program planners in developing effective strategies to reduce mortality among HIV-TB Co-infected patients in Ethiopia. Additionally, it will serve as a baseline data for future researchers.

## Methods

### Protocol registration and reporting

This systematic review and meta-analysis have been registered in the International Prospective Register of Systematic Reviews (PROSPERO) with registration (ID: CRD42023466558), which is available at https://www.crd.york.ac.uk/prospero/#myprospero. The Preferred Items for Systematic Review and Meta-Analysis (PRISMA) checklist was used for reporting this review [22] (S1 Table).

### Databases and search strategy

A literature search through three databases; PubMed/Medline, CINAHL, and Global Index Medicus (GIM) with Medical Subject Heading (MeSH) terms and keywords was done. In addition, Gray literature was searched on Google, Google Scholar, and Worldwide Science. Furthermore, articles from reference lists of included studies, and related reviews were assessed, and retrieved. We searched for articles until November 25, 2023. The search was focused on studies that reported the proportion of mortality and its predictors in Ethiopia. For searching literature the following keywords and phrases were used: "Magnitude", "Occurrence", "Prevalence", "Incidence" "Burden" "Proportion", "Epidemiology", "Mortality", "Death", "Survival status", "Survival rate", "HIV-TB co-infected patients" "HIV-TB co-infected cases", "HIV-TB co-infection", "HIV and TB co-infection", "Tuberculosis/HIV co-infected patients", "MTB/HIV co-infection", "Tuberculosis", "TB", "Human immunodeficiency virus", "HIV/AIDS", "TB/HIV", "HIV TB co-infected patients", "Predictors", "determinants" "Associated factors", "Risk factors". The search strings were formed using the "AND" "OR", and "Asterisk" Boolean operators. Three authors (WTW, BTR, and CAW) systematically searched articles published in the English language (S2 Table).

## Eligibility criteria

Studies that fulfilled the following major criteria were considered for inclusion in this systematic review and meta-analysis.

## Inclusion criteria

This study included all published observational studies (i.e. cross-sectional, and cohort) conducted in Ethiopia, that reported mortality and/or predictors among HIV-TB co-infected patients. Included studies were in the Population, Exposure, Comparison, and Outcome (PECO) framework (P: HIV-TB coinfected patient, E: HIV-TB coinfection, E: patients with no HIV-TB co-infection and O: mortality among HIV- TB co-infected patients from any other causes). On the other hand, books, meeting reports, and studies that don't contain the necessary data were excluded (**S3 Table**).

*Screening and data extraction.* Based on the pre-specified inclusion criteria, two reviewers (WTW and BTR) screened the titles and abstracts of the studies. They then conducted an independent assessment by reviewing the full text of the studies, and extracted the data from November 26, 2023 to December 26,2023. Any discrepancies between the two reviewers were solved through discussion and common consensus. The following information were extracted: first author's name, year of publication, study region & design, population, sample size, follow-up period, proportion of mortality, and the hazard ratio for the predictors (**S1 File**).

## Quality assessment

For this review, Endnote version X9 (Thomson Reuters, Philadelphia, USA) software was used to remove duplicate studies. Three investigators (WTW, BTR & CAW) assessed the quality of eligible articles using the Joanna Briggs Institute (JBI) Critical appraisal tools designed for cross-sectional and cohort studies [23]. The cohort checklist includes 11 indicators, while the cross-sectional study consists of 8 indicators. These indicators were converted to a percentage scale, with quality scores graded as high if >80%, medium if between 60 and 80%, and low, if < 60%. Any inconsistencies or discrepancies were resolved through discussion with the fourth and fifth authors (ABZ &BTL) (**S4 Table**).

## Measurement of outcome variables

The present systematic review and meta-analysis had two outcomes, the first was mortality from any cause in patients with HIV-TB co-infected, and the second outcome was predictors of mortality. The pooled hazard ratio (PHR) with 95% confidence intervals was calculated to evaluate these predictors of mortality.

## Data synthesis and statistical analysis

Microsoft Excel was used to extract the data and the extracted data were imported into STATA version 18 for analysis. To estimate the proportion of mortality and the effect size of predictors among HIV-TB co-infected patients, a random effect model using the Restricted Maximum Likelihood (REML) method was used [24]. Statistical heterogeneity across studies was assessed using the Cochrane Q-test, tau-squared, and $I^2$ –statistics. $I^2$-statistic of 25, 50, and 75% represent low, moderate, and high heterogeneity respectively [25]. The variation in the proportion of mortality was adjusted through subgroup analysis based on the study characteristics, including study region and age group. Additionally, a sensitivity analysis was conducted to evaluate the impact of each study on the overall estimate. Furthermore, publication bias was assessed through visual inspection of the funnel plot, and Egger's test [26]. Results of Egger's test (P-

value ≤ 0.05) are considered indicative of publication bias (small study effect). This was addressed using a non-parametric trim and fill analysis with a random effects model [27]. The trim and fill funnel plot analysis was confirmed by a Galbraith plot, and the studies lying inside and outside the central 95% pseudo-CI region agreed with the distributional features in the pooled mortality forest plot. The results were presented using forest plots and tables.

## Results

### Review processes and findings

In our search, we retrieved a total of 886 studies. Out of these, 807 studies were obtained from three data databases: PubMed (623), Global Index Medicus (159), and CINAHL (25). We removed 123 duplicates, and 684 studies remained in Endnote, which were screened by title and abstract. After this screening, 624 irrelevant studies were excluded. Subsequently, 60 studies were assessed by full-text review. Based on our criteria, 35 studies were excluded and 25 studies were included in the final meta-analysis. In addition to databases, we utilized other relevant sources, such as free web search engines (Google, Google Scholar, Worldwide Science), which yielded 79 full-text articles. From these, 67 irrelevant studies were excluded, and 12 articles were included. In total, 37 studies were ultimately included (**Fig 1**).

### Characteristics of included studies

In the current study, a total of 37 studies [20, 21, 28–62], with a sample size of 12,061 participants were included. Among these, 8 studies were conducted in the Amhara region [29, 32, 33, 36, 46, 52, 54, 61], 7 in the Oromia region [30, 35, 39, 43, 55, 58, 60], 8 in South Nation and Nationality People Region (SNNPR) [20, 34, 38, 41, 45, 51, 53, 62], 5 in Tigray region [21, 31, 37, 56, 59], 3 in Addis Ababa [40, 42, 44], 1 in Dire Dewa [57], 1 in Harari [49], 2 in Harari

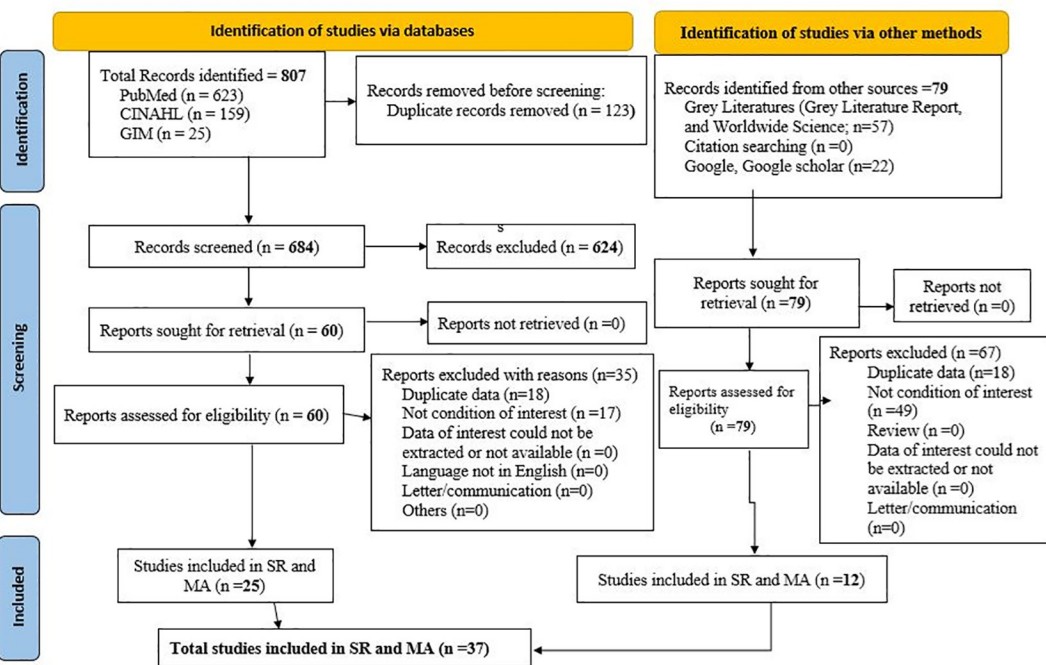

**Fig 1. PRISMA 2020 flow diagram for study selection of mortality and predictors among HIV-TB co-infected patients in Ethiopia.**

**Table 1. Characteristics of included studies on mortality and predictors among HIV-TB co-infected patients in Ethiopia.**

| ID | Author, Year of Publication | Region | Study design | Follow-up Period | Age group | Sample size | Mortality% |
|---|---|---|---|---|---|---|---|
| 1 | Dawit, et al, 2021 | SNNPR | Retrospective Cohort study | 2009–2018 | Pediatrics | 274 | 17.15 |
| 2 | Gemechu, et al,2022 | SNNPR | Retrospective Cohort study | 2009–2019 | Pediatrics | 284 | 12.32 |
| 3 | Atallel, et al 2018 | Amhara | Retrospective Cohort study | 2005–2017 | Pediatrics | 271 | 14.02 |
| 4 | Chanie, et al, 2021 | Amhara | Retrospective Cohort study | 2014–2021 | Pediatrics | 227 | 17.18 |
| 5 | Negussie, et al,2021 | Tigray | Retrospective Cohort study | 2008–2018 | Pediatrics | 253 | 15.01 |
| 6 | Alula, et al, 2017 | Ethiopia | Longitudinal study | 2005–2017 | Both | 355 | 15.49 |
| 7 | Shaweno, et al, 2012 | SNNPR | Retrospective Cohort study | 2006–2010 | Adult | 370 | 13.50 |
| 8 | Gebreyes, 2023 | Amhara | Retrospective Cohort study | 2005–2016 | Adult | 407 | 29.48 |
| 9 | W/Gebreal, et al, 2018 | Harari and Dire Dawa | Retrospective cross-sectional | 2008–2014 | Adult | 627 | 8.60 |
| 10 | Habtamu, et al, 2021 | Tigray | Retrospective cohort study | 2011–2015 | Adult | 210 | 80.48 |
| 11 | Abrha, et al, 2015 | Oromia | Retrospective cohort study | 2010–2012 | Adult | 272 | 20.22 |
| 12 | Birhan, et al, 2021 | Amhara | Retrospective Follow-up study | 2014–2019 | Adult | 243 | 35.39 |
| 13 | Refera, et al, 2013 | Oromia | Retrospective cohort study | 2006–2012 | Adult | 501 | 15.77 |
| 14 | Gezae, et al, 2019 | Tigray | Retrospective follow-up study | 2009–2016 | Adult | 305 | 23.00 |
| 15 | Lelisho, et al, 2022 | Oromia | Retrospective cohort study | 2014–2022 | Adult | 402 | 20.90 |
| 16 | Sileshi, et al, 2013 | Amhara | Retrospective cohort study | 2009–2012 | Adult | 422 | 22.03 |
| 17 | Belayneh, et al, 2015 | Tigray | Cross-sectional | 2009–2011 | Adult | 342 | 25.73 |
| 18 | Ali, et al, 2016 | Ethiopia | Cross-Sectional study | 2013–2013 | Adult | 169 | 8.29 |
| 19 | Beyen, et al 2016 | Amhara | Historical cohort study | 2006–2010 | Both | 629 | 17.98 |
| 20 | G/Mariam, et al, 2016 | Oromia | Retrospective document review | 2008–2014 | Both | 156 | 16.66 |
| 21 | Reepalu, et al, 2017 | Oromia | Prospective cohort study | 2011–2013 | Adult | 141 | 9.00 |
| 22 | Ifa, 2018 | SNNPR | Retrospective cross-sectional study | 2012–2016 | Both | 159 | 4.40 |
| 23 | Tola, et al,2019 | Harari and Dire Dawa | Retrospective Cross-sectional study | 2012–2017 | Both | 349 | 7.69 |
| 24 | Geliso, et al,2020 | Tigray | Retrospective cohort study | 2016–2017 | Adult | 106 | 15.10 |
| 25 | Wondimu, et al, 2020 | SNNPR | Retrospective Cohort study | 2007–2017 | Adult | 364 | 22.80 |
| 26 | Sime, et al. 2022 | Harari and Dire Dawa | Retrospective Cohort study | 2014–2018 | Both | 263 | 22.81 |
| 27 | Seyoum, et al, 2022 | Addis Ababa | Retrospective Cohort study | 2011–2018 | Adult | 1123 | 4.45 |
| 28 | Fekadu, et al, 2022 | Oromia | Retrospective Cohort study | 2013–2019 | Adult | 124 | 19.36 |
| 29 | Alemu, et al, 2021 | SNNPR | Retrospective cohort study | 2015–2019 | Both | 302 | 5.63 |
| 30 | Teshome, et al, 2017 | SNNPR | Retrospective Cross-sectional study | 2012–2015 | Adult | 188 | 12.80 |
| 31 | Palme, et al,2002 | Addis Ababa | Prospective cohort study | 1995–1997 | Pediatrics | 58 | 37.93 |
| 32 | Sinshaw, et al 2017 | Amhara | Cross-Sectional study | 2010–2016 | Both | 308 | 10.07 |
| 33 | Balcha, et al, 2015 | Oromia | Prospective cohort study | 2010–2013 | Adult | 439 | 6.15 |
| 34 | Adegeh, et al, 2021 | Amhara | Retrospective Cohort study | 2007–2012 | Adult | 314 | 21.02 |
| 35 | H/Giorgis, et al.2018 | Harari and Dire Dawa | Retrospective Cohort Study | 2012–2016 | Both | 471 | 16.80 |
| 36 | Kassa, et al, 2012 | Addis Ababa | Retrospective Cohort Study | 2005–2009 | Adult | 270 | 21.11 |
| 37 | Lelisho, et al,2023 | SNNPR | Retrospective Cohort Study | 2015–2020 | Adult | 363 | 21.76 |

*SNNPR = South Nation and Nationality People Region.

and Dire Dawa [47, 50], and 2 across various parts of Ethiopia [28, 48]. The studies comprised 27 cohort studies, 9 cross-sectional studies, and 1 longitudinal study. The minimum and maximum sample sizes of the included studies were 58 [42], and 1123 [44] respectively (**Table 1**)

## Quality of included studies

In this study, we found 27 Cohort studies, 9 cross-sectional studies, and 1 longitudinal. The JBI quality appraisal checklist for cohort and cross-sectional studies was utilized to evaluate

the quality of the included studies. The result of the quality assessment ranged from 81.81 to 100 for both study designs (**S4 Table**).

## Pooled proportion of mortality

In this study, to estimate the pooled proportion of mortality, 37 studies were included. The death rate varied from 4.4% in a study conducted in SNNPR [20] to 80.45% in a study conducted in Tigray [21]. The present meta-analysis, employing a random effect model using the Restricted Maximum Likelihood (REML) method showed that the pooled proportion of mortality among HIV-TB co.-infected patients in Ethiopia was 18.42% (95% CI: 14.27–22.57). In this study, significant heterogeneity was observed within studies with a Q = 123.16, p-value = 0.00), Tau-squared = 160.41, H = 50.46 $I^2$ = 98.02%, P-value = 0.00. The adjusted prediction interval was found to range from 0 to 80%) (**Fig 2**).

## Publication bias and its handling mechanism

An Egger's test value β = 7.11, P = 0.001, indicated a significant publication bias, and highly non-normal distribution. A funnel plot confirmed both the bias and the non-normal distribution. Accordingly, the non-parametric trim and fill analysis was done, and three studies were imputed. However, the non-parametric trim and fill analysis failed to resolve the bias because 22 of the 37 studies and the 3 imputed studies remained outside the pseudo 95% CI (**Fig 3**). A Galbraith plot confirmed the results of the funnel plot. The overall pooled proportion of mortality in the non-parametric trim and fill analysis was 15.66% (95% CI:10.38–20.957).

## Handling heterogeneity

In this study, there was high heterogeneity (Q = 123.16, P = 0.00, Tau-squared = 160.41, $H^2$ = 50.46 $I^2$ = 98.02%, P-value = 0.00) between and within the 37 studies, and a forest plot was done using the REML method to show the pooled mortality for the 37 studies (**Fig 2**). In addition, a Galbraith plot was done, and it showed some studies fell outside the 95% CI, which showed the presence of statistically significant heterogeneity (**Fig 4**). Although the pooled mortality in the forest plot was 18.42% (95% CI; 14.27–22.57), there was a significant range in mortality values for Ifa 2018, 4.40% (95% CI: 1.21–7.59) [20], to Habtamu et.al. 2021, 80.48% (95% CI: 75.12–85.84) [21]. Due to this high heterogeneity, subgroup and sensitivity analyses were performed by region (**Fig 5**), and by age group (**Fig 6**).

## Subgroup analysis

Sub-group analysis was performed based on study region, and age group of the study populations. The highest proportion of mortality was reported in Tigray region 31.86% (95% CI: 7.69–56.03), followed by Amhara region 20.73% (95% CI: 15.15–26.30), Addis Ababa 20.36% (95% CI: 1.76–38.97), Oromia 15.19% (95% CI: 10.73–19.65), SNNPR 13.66% (95% CI: 8.98–18.35), Harari and Dire Dawa 13.74% (95% CI: 6.86–20.62), and in studies conducted across various parts of Ethiopia 11.95% (95% CI: 4.91–19.00). In this sub-group analysis, there was variability within groups (P = 0.03), and in the overall pooled estimate (I = 98.02, P-value = 0.00). However, no heterogeneity was found between groups (regions) as indicated by (test of group differences: Q (6) = 7.33, P = 0.30) (**Fig 5**), (**Table 2**).

Subgroup analysis was also by age group of the patient. The incorporated studies categorize age groups into pediatrics (under 15 years), adults (15 years and older), and both (studies that include both adults and pediatrics). The highest mortality rate was observed among adults, with a rate of 20.70% (95% CI: 14.29–27.11). In this sub-group analysis, there was significant

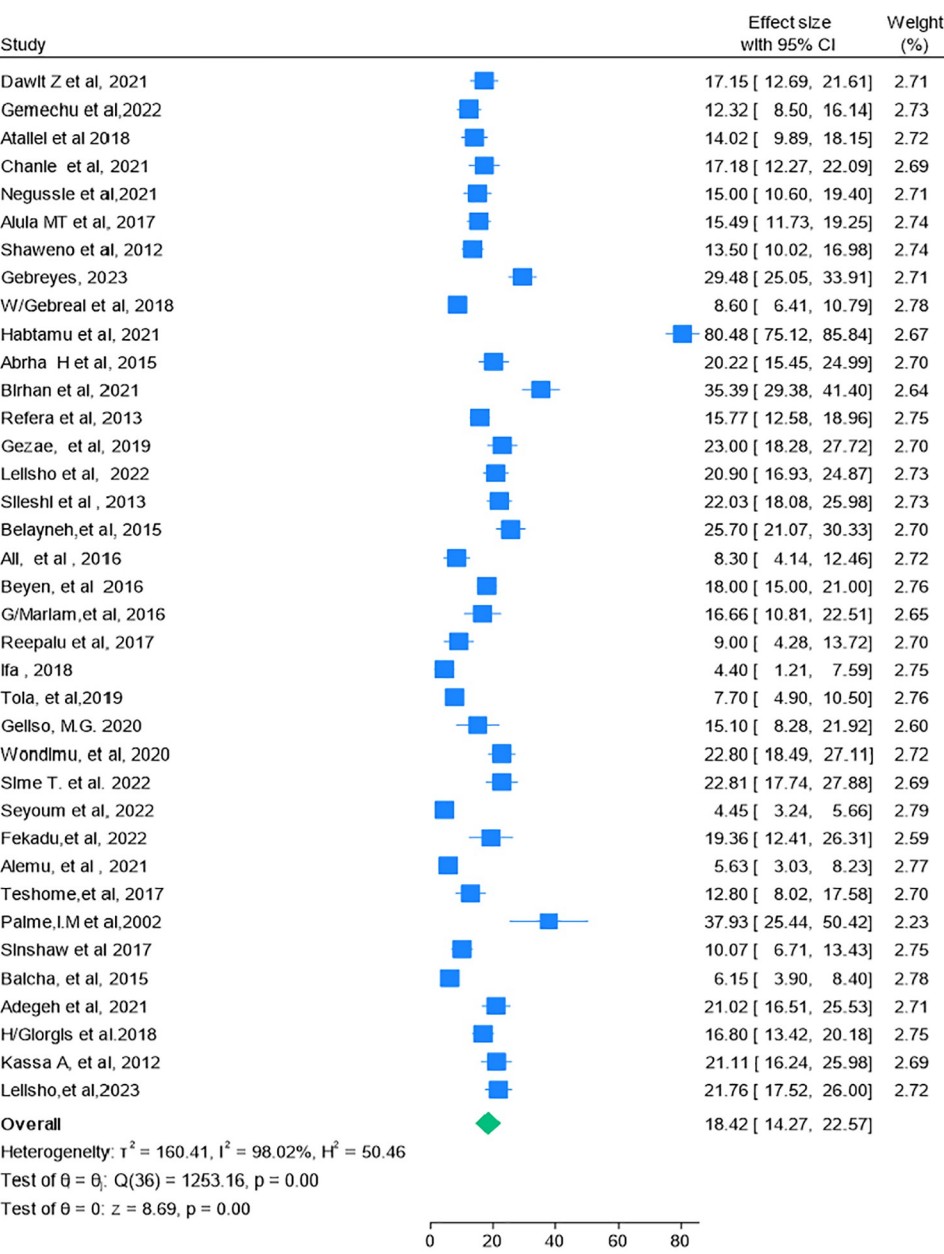

**Fig 2. Forest plot showing the magnitude of mortality among HIV-TB co-infected patients in Ethiopia.**

heterogeneity within groups (P = 0.00), and in the overall pooled estimate (P = 0.00). However, there was no heterogeneity between groups (age groups) as indicated by (the test of group differences: ($Q_b$ (2) = 4.39, P = 0.11) (**Fig 6**).

## Sensitivity analysis

A sensitivity analysis was conducted using a random effect model for the overall mortality rate In this analysis, there were no influential studies, as all of the single estimates of the leave-one-out analysis fell within the confidence interval of the pooled proportion of mortality18.42%

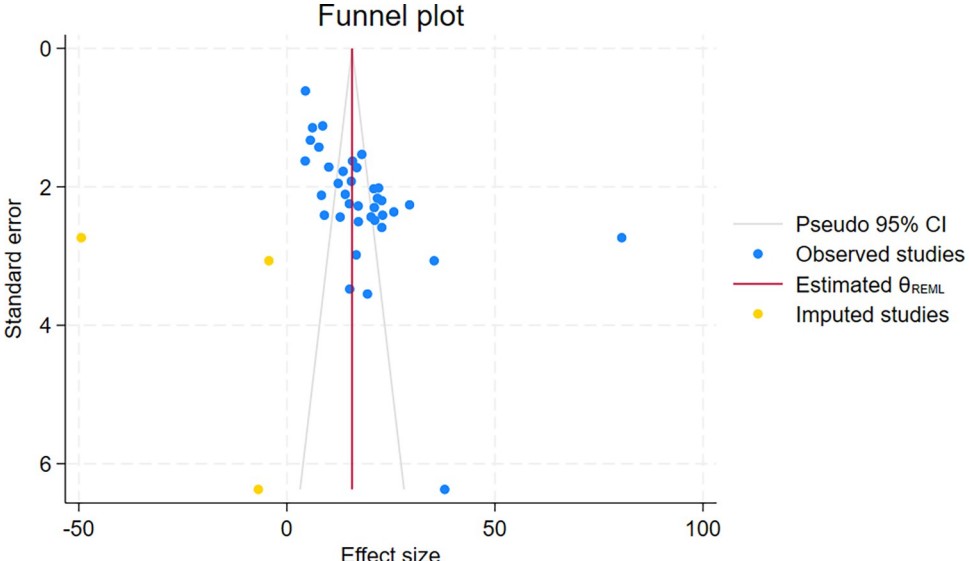

**Fig 3. Non-parametric funnel plot with trim and fill analysis showing publication bias to assess the pooled mortality among HIV-TB co-infected patients.**

(95% CI:14.27–22.57). Additionally, the weights in Fig 2 are approximately equal and group around 100/ 37 = 2.70, which is satisfactory, and shows that no study unduly influences the overall weighting. These results indicate that no single study significantly affects the overall magnitude of mortality (**S1 Fig**).

## Predictors of mortality

In this study, the pooled effect size of the 20 independent predictors was assessed. The evaluated predictors include sex, anemia, CD4 count, not taking CPT, not taking Isoniazid Preventive Therapy (IPT), risky behavior, extrapulmonary tuberculosis, residence, poor adherence,

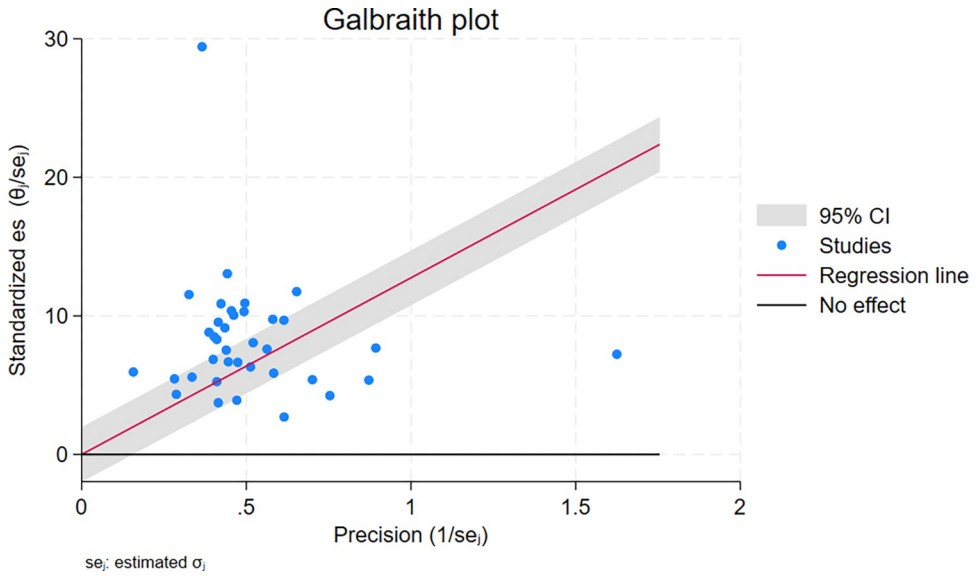

https://doi.org/

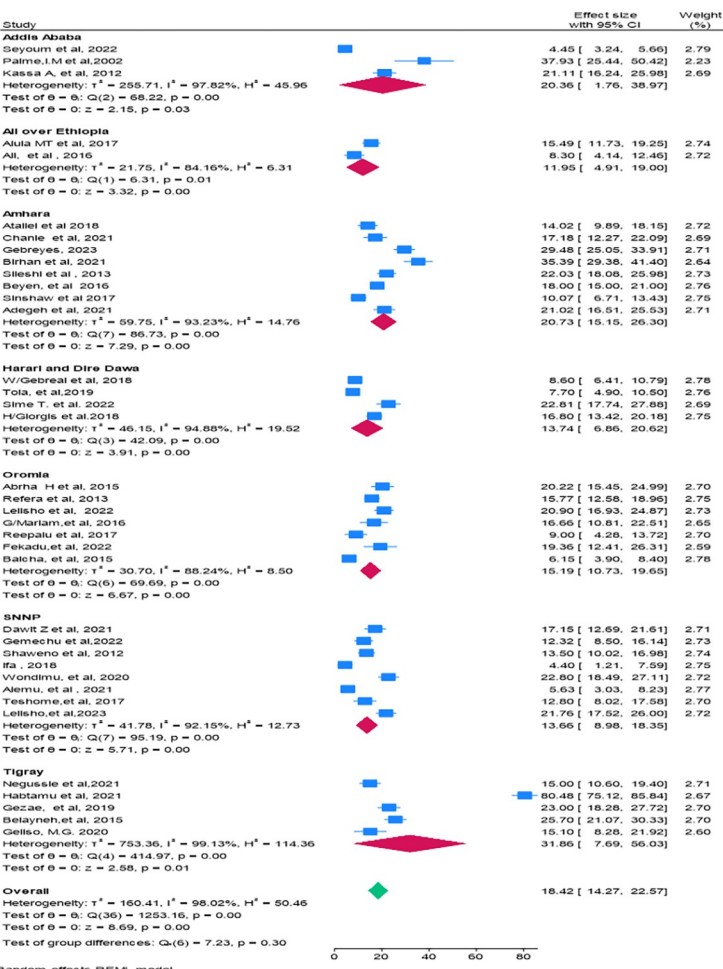

**Fig 5. Forest plot showing subgroup analysis of the proportion of mortality among HIV-TB co-infected patients by region.**

non-disclosure, ART failure, Multi-Drug Resistance TB (MDR) TB, presence of opportunistic infection, WHO HIV-stages, commercial sex workers, functional status (bedridden vs. ambulatory). Finally, as shown below, only four predictors anaemia, poor adherence, not taking CPT, and extrapulmonary tuberculosis were found to be significant predictors of mortality in the random effect model.

## Anaemia

Among thirty-seven studies, six studies [29, 33, 34, 38, 57, 59] were included to examine the association between anaemia and mortality. The pooled hazard ratio was found to be 2.25 (95% CI: 1.65–3.07). This meta-analysis showed that the hazard of death among anaemic patients was 2.25 times higher compared to non-anaemic patients. The heterogeneity test showed no evidence of variation across studies ($I^2 = 0.0$), (**Fig 7**).

## Poor adherence

Six studies [29, 33, 34, 38, 54, 59] assessed the association between poor adherence to ART and mortality among HIV–TB co-infected patients. The pooled hazard ratio was 2.42(95% CI: 1.39–4.21). Therefore, this meta-analysis revealed that the hazard of death among co-infected

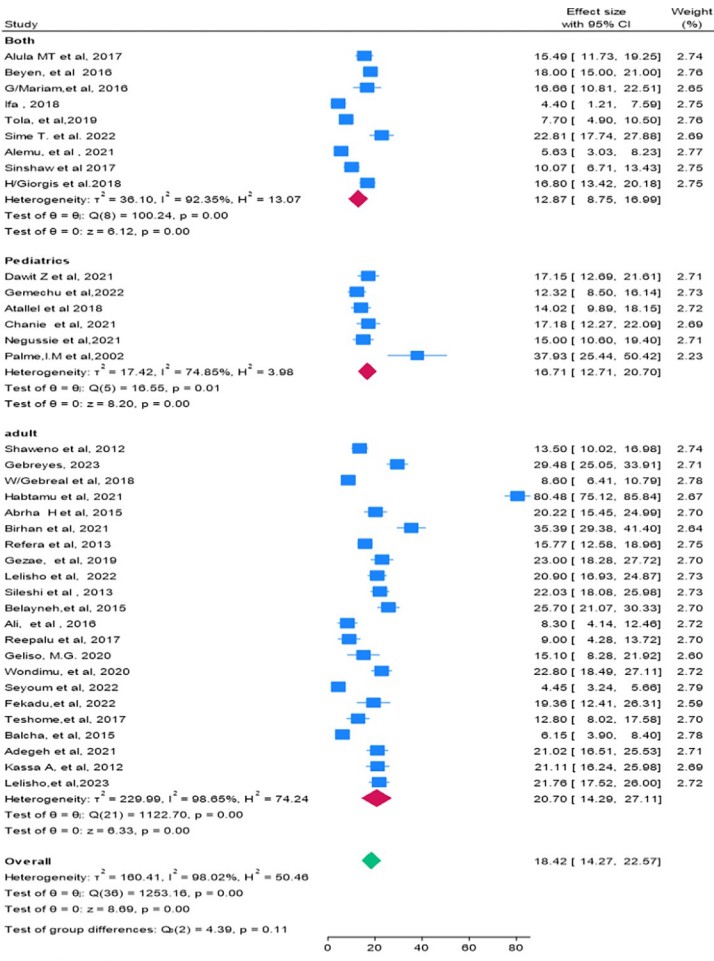

**Fig 6. Forest plot for subgroup analysis of the proportion of mortality among HIV-TB co-infected patients by age group.**

individuals with poor adherence was 2.42 higher compared to those with good adherence. For this predictor publication bias was not performed due to a small number of studies (**Fig 8**).

## Not taking CPT

In this review, 10 studies [29, 33, 34, 41, 46, 51, 56–59] reported the association between not taking Co-trimoxazole Prophylaxis Therapy (CPT) and mortality among HIV-TB co-infected

**Table 2. Sub-group analysis of the proportion of mortality among HIV-TB co-infected patients by region.**

| Region | No of studies | Proportion (95%CI) | Q-value | $I^2$ | P-value |
|---|---|---|---|---|---|
| Addis Ababa | 3 | 20.36 (1.76–38.97) | 68.22 | 97.82 | 0.03 |
| Amhara | 8 | 20.73 (15.15–26.30) | 86.73 | 93.23 | 0.00 |
| All over Ethiopia | 2 | 11.95 (4.91–19.00) | 6.31 | 84.16 | 0.00 |
| Harari and Dire Dawa | 4 | 13.74 (6.86–20.62) | 42.09 | 94.88 | 0.00 |
| Oromia | 7 | 15.19 (10.73–19.65) | 69.69 | 88.24 | 0.00 |
| SNNP | 8 | 13.66 (8.98–18.35) | 95.19 | 92.15 | 0.00 |
| Tigray | 5 | 31.86 (7.69–56.03) | 414.97 | 99.13 | 0.01 |
| Overall | 37 | 18.42 (14.27–22.57) | 1253.16 | 98.02 | 0.00 |

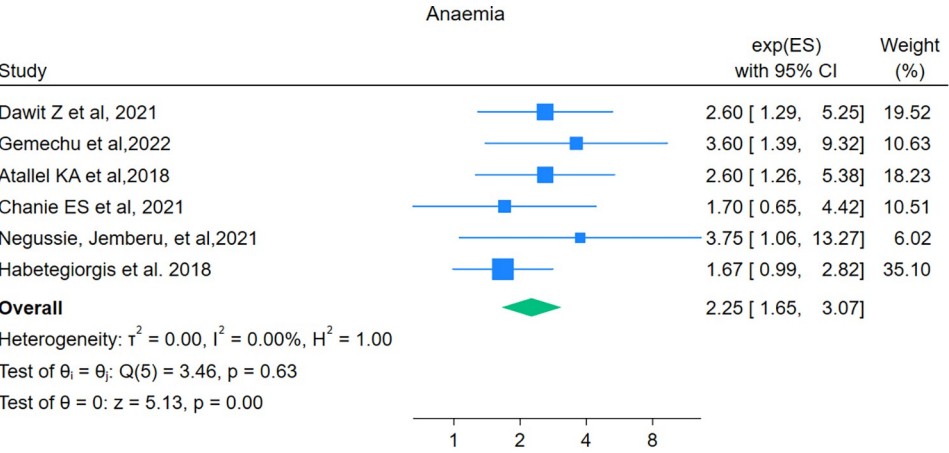

**Fig 7. Forest plot showing the pooled hazard ratio of the association between anaemia and mortality among HIV-TB co-infected patients.**

patients. The pooled hazard ratio was found to be 1.87 (95% CI: 1.28–2.73). The current meta-analysis showed that patients who were not taking CPT were nearly two times at higher risk of death compared with their counterparts. The Eggers test showed that there was no publication bias (beta = 1.23, P = 0.58) (**Fig 9**).

## Extrapulmonary tuberculosis

In this review, 11 studies [29, 34, 36, 38, 39, 46, 52, 54, 56, 59, 60] reported the association between Extra Pulmonary Tuberculosis (EPT), and mortality among HIV-TB co-infected patients. In this meta-analysis, the pooled hazard ratio was 1.23(95% CI: 1.01–1.51), which means patients who had EPT had a 1.23 times higher hazard of death compared with their counterparts. Likewise, to the above predictors, there was no statistically significant publication bias (Egger test P = 0.244) (**Fig 10**).

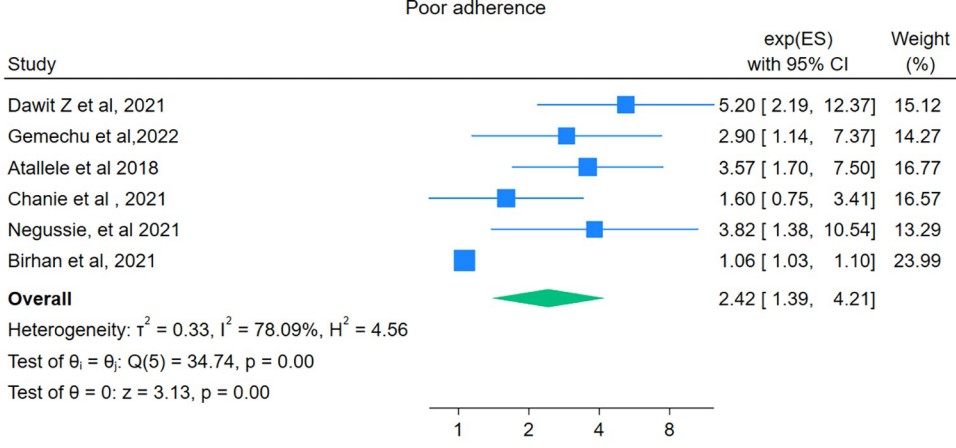

**Fig 8. Forest plot showing the pooled hazard ratio of the association between poor adherence and mortality among HIV-TB co-infected patients.**

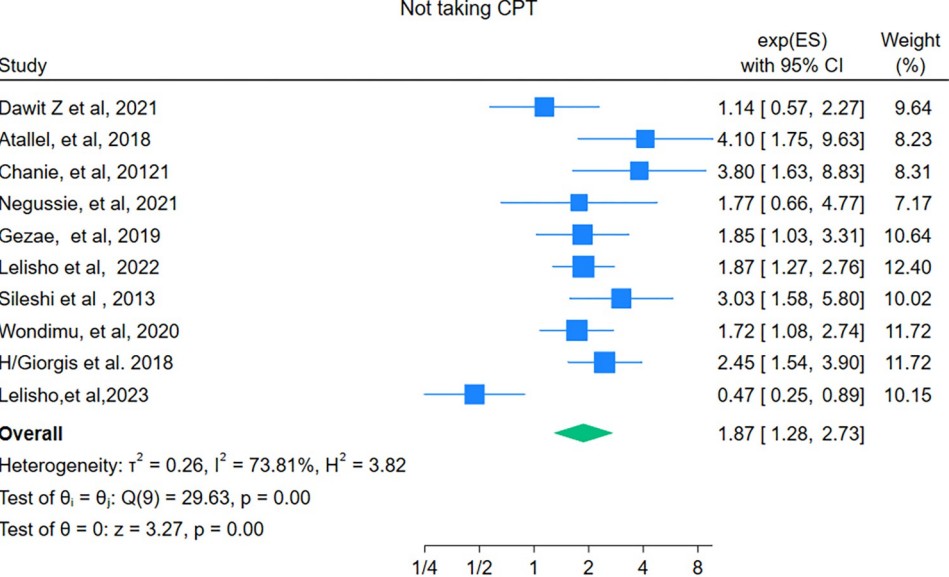

**Fig 9. Forest plot showing the pooled hazard ratio of the association between not taking CPT and mortality among HIV-TB co-infected patients.**

## Discussion

This systematic review and meta-analysis aimed to provide and synthesize evidence regarding mortality and its predictors among HIV-TB co-infected patients in Ethiopia. This study showed that the national level magnitude of mortality among HIV-TB co-infected patients was 18.42% (95% CI: 14.27–22.57).

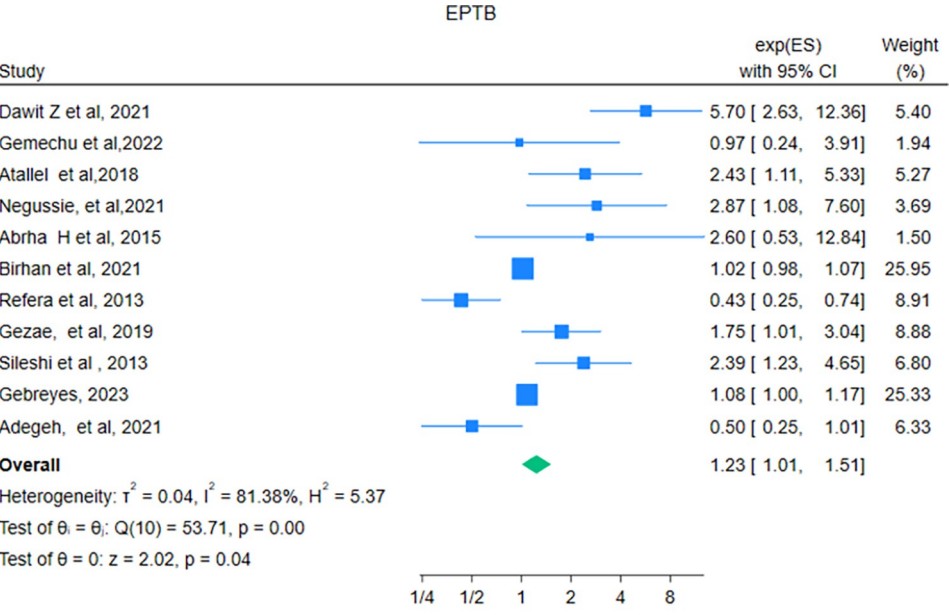

**Fig 10. Forest plot showing the pooled hazard ratio of the association between extrapulmonary tuberculosis and mortality among HIV-TB co-infected patients.**

This finding aligns with studies conducted in Sub-Saharan Africa (SSA) (18.1%) [1], Europe and Latin America (19%) [63], and a global systematic review and meta-analysis (15%) [64]. However, this finding was lower than that of a study conducted in Eastern Europe (29%) [63], Brazil (27.4%) [65], and another global systematic review and meta-analysis (33.56%) [66]. This discrepancy may be attributed to geographical variation and the presence of some challenging comorbidities. For example, in those studies, a high number of patients had drug-resistant tuberculosis, whereas in the current study setting the prevalence of multi-drug resistance tuberculosis is low. The other possible justification is the study period, those studies were incorporated studies conducted before the commencement of the test-and-treat strategy, while in the present study, most of the included studies were conducted after the implementation of test and treat all strategy in Ethiopia since 2017, which is a proven strategy in reducing mortality [7, 67].

On the other hand, the finding of the present study was higher than a study conducted in Western Europe (4%) [9, 63], Latin America (11%) [63], United Kingdom (4%), the Caribbean, Central and South America, Central, East, and West Africa (12%) [13], studies in Brazil (13%) [9] & (5.9%) [68], and a global systematic review and meta-analysis (11%) [9]. This discrepancy might be related to limited access to appropriate therapy, delayed diagnosis, and poor awareness of these diseases. In low-income countries like Ethiopia, due to the unavailability/ inadequate supply of diagnostic tools and antiretroviral/anti-TB services, and poor access to healthcare facilities, there is poor management of HIV-TB co-infected patients, and this ultimately leads to increased mortality [69]. On the other hand, delay in ART and Anti-TB initiation resulted in advanced clinical stage, opportunistic infection, and immune reconstitution inflammatory syndrome. Before WHO launched the immediate initiation of ART without clinical or immunological status, there was a delayed initiation of therapy in Ethiopia which escalated the mortality rate [70].

Due to heterogeneity, subgroup analysis was conducted based on regions. Consequently, the highest mortality rate was reported in the Tigray region at 31.86%, followed by the Amhara region 20.73%, while, the lowest mortality rate was observed in studies conducted in all parts of Ethiopia at 11.95%. The possible reason for this discrepancy might be poor awareness about those infectious diseases. Due to this, the co-infected individuals may have poor adherence to medications, and poor access to ART/anti-TB, which may result in increased mortality. Besides, the national HIV/AIDS and TB control program might not be uniformly implemented across different regions of Ethiopia, leading to variation in the survival of co-infected patients. Additionally, TB patients in the Tigray region have poor treatment outcomes [71], which may further elevate the mortality rate in case of co-infection. Therefore, supervision of HIV and TB control services is significantly needed to harmonize the service provision. Moreover, it might be due to demographic factors, like, lifestyle/awareness and socioeconomic factors which may further influence the mortality of co-infected patients. In addition to regions, sub-group analysis was done based on the participants' age group. Accordingly, the highest proportion of mortality was found among adult patients. This might be due to increased occupational and social interactions in adults. Factors such as unprotected sexual intercourse, injection drug use, and occupational and environmental exposure to HIV, and TB, ultimately resulted in mortality [72]. On the other hand, adults may have other underlying medical problems like diabetes, cardiovascular diseases, etc., which further weaken their immune system, and decrease the survival of HIV-TB co-infected patients.

The present study showed that patients presented with anaemia were more than two times at hazard of death compared with their counterparts. This finding was congruent with systematic review and meta-analysis conducted globally [73], and South Africa [74]. This might be due to both HIV and TB infection compromising the immune system of the patients and

affecting lung function, and anaemia exacerbates this problem of oxygen supply, which can result in hypoxia in vital organs such as the brain and heart, this leads to morality [7, 75]. On the other hand, anaemia increases the risk of susceptibility to opportunistic infection and disease progression, which increases the hazard of death [7, 76]. Besides this, in those individuals' anaemia may be due to nutritional deficiency, which is a significant cause of mortality in co-infected patients [7].

The current study also revealed that patients who had poor adherence to ART were above twofold at hazard of death compared with those individuals with good adherence. A similar finding was reported in a study conducted in Spain [77]. This might be due to, poor adherence could result in reduced viral load suppression and decreased CD4 cell count, which leads to uncontrolled HIV replication, and the patient's immune system becomes compromised and can't fight off opportunistic infections, which ends up in mortality. The second reason could be poor adherence causes drug resistance, treatment failure, and susceptibility to opportunistic infection, hence ultimately it results in increased mortality. Thus, healthcare providers should emphasize, and provide counseling about the importance of good adherence to ensure optimal treatment outcomes [7].

In this study, patients who were not receiving co-trimoxazole preventive therapy had twice the hazard of death compared with their counterparts. This increased risk may be due to non-adherence to Co-trimoxazole therapy in individuals co-infected with HIV and TB, which raises the likelihood of opportunistic infections, particularly pneumocystis jirovecii pneumonia, and impairs immune recovery, ultimately leading to mortality [19]. In addition to the above predictors, extrapulmonary tuberculosis was an independent predictor of mortality in HIV-TB co-infected patients. This finding was consistent with a study conducted in South Africa [78]. This is due to the fact, that extrapulmonary tuberculosis further compromises, the immune system's ability to fight off infections. This leads to uncontrolled dissemination of TB throughout the body and increased viral load, which results in a decreased survival rate. On the other hand, extra pulmonary tuberculosis may involve critical organs such as brain, spine, and meninges which leads to life threatening complications and increased mortality rate [7, 19].

## Strength and limitations

This study tried to show the pooled magnitude of mortality among HIV-TB co-infected patients at a national level for the first time. However, our study had the following limitations: the significant heterogeneity in the pooled proportion of mortality of the 37 studies, remained in the subgroup and sensitivity analysis and within study analysis. The funnel plot and Galbraith plot analysis indicated significant bias which was not resolved in the fill and trim procedure in the case of the funnel plot. Nevertheless, the paper has significant value in establishing benchmark values in the case of a country with significant diversity in its independent predictors. Additionally, we could not analyze certain factors, such as nutritional status due to inconsistent measurements.

## Conclusion

This study indicated that, at the national level, 18.42% of patients co-infected with HIV and Tuberculosis (TB) died. Anaemia, poor adherence, not receiving CPT, and extrapulmonary tuberculosis were significant predictors of mortality. The high mortality of HIV-TB co-infected patients is an urgent concern. Hence, the FMOH of Ethiopia needs to expand and strengthen the integration of HIV-TB collaborative services and consistent monitoring activities across all health facilities of the country, with critical attention for patients presented with the aforementioned predictors.

## Supporting information

**S1 Table. PRISMA 2020 checklist.**
(DOCX)

**S2 Table. Search strategies and terms in different databases.**
(DOCX)

**S3 Table. List of all identified studies in the literature search including those excluded studies from the analyses.**
(DOCX)

**S4 Table. Quality assessment JBI criteria.**
(DOCX)

**S1 File. Extracted dataset of the included studies.**
(XLSX)

**S1 Fig. Forest plot showing sensitivity analysis of the magnitude of mortality among HIV-TB co-infected patients in Ethiopia.**
(TIF)

## Acknowledgments

We authors would like to thank Ambo University for providing us with internet services to access available databases and web pages.

## Author Contributions

**Conceptualization:** Wubet Tazeb Wondie, Chalachew Adugna Wubneh, Belay Tafa Regassa.

**Data curation:** Wubet Tazeb Wondie, Chalachew Adugna Wubneh, Bruck Tesfaye Legesse, Belay Tafa Regassa.

**Formal analysis:** Wubet Tazeb Wondie, Chalachew Adugna Wubneh, Gebrehiwot Berie Mekonen, Alemu Birara Zemariam, Zenebe Abebe Gebreegziabher, Belay Tafa Regassa.

**Funding acquisition:** Gezahagn Demsu Gedefaw.

**Investigation:** Wubet Tazeb Wondie.

**Methodology:** Wubet Tazeb Wondie, Chalachew Adugna Wubneh, Bruck Tesfaye Legesse, Gebrehiwot Berie Mekonen, Alemu Birara Zemariam, Zenebe Abebe Gebreegziabher, Gemechu Gelan Bekele, Belay Tafa Regassa.

**Project administration:** Wubet Tazeb Wondie, Gezahagn Demsu Gedefaw.

**Resources:** Belay Tafa Regassa.

**Software:** Wubet Tazeb Wondie, Gebrehiwot Berie Mekonen, Alemu Birara Zemariam.

**Supervision:** Chalachew Adugna Wubneh, Bruck Tesfaye Legesse, Gebrehiwot Berie Mekonen, Alemu Birara Zemariam, Zenebe Abebe Gebreegziabher, Gezahagn Demsu Gedefaw, Gemechu Gelan Bekele, Belay Tafa Regassa.

**Validation:** Wubet Tazeb Wondie, Chalachew Adugna Wubneh, Bruck Tesfaye Legesse, Alemu Birara Zemariam, Belay Tafa Regassa.

**Visualization:** Wubet Tazeb Wondie, Chalachew Adugna Wubneh, Bruck Tesfaye Legesse, Zenebe Abebe Gebreegziabher, Gemechu Gelan Bekele, Belay Tafa Regassa.

**Writing – original draft:** Chalachew Adugna Wubneh.

**Writing – review & editing:** Wubet Tazeb Wondie, Bruck Tesfaye Legesse, Gebrehiwot Berie Mekonen, Alemu Birara Zemariam, Zenebe Abebe Gebreegziabher, Gezahagn Demsu Gedefaw, Gemechu Gelan Bekele, Belay Tafa Regassa.

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
