## [Decision Letter · Decision Letter 0]

12 Jun 2024

PONE-D-24-06919Mortality and predictors among HIV/TB co-infected patients in Ethiopia: A Systematic Review and Meta-analysis.PLOS ONE

Dear Dr. Wondie,

Thank you for submitting your manuscript to PLOS ONE. After careful consideration, we feel that it has merit but does not fully meet PLOS ONE’s publication criteria as it currently stands. Therefore, we invite you to submit a revised version of the manuscript that addresses the points raised during the review process.

 This paper was reviewed by an expert statistician who has made comprehensive comments on the statistical methodology including the construction of the funnel plots with respect to the presence of outliers and the expression of p values. The reviewers also expressed concern with the selection of the MeSH terms which must be fully explained. Please submit your revised manuscript by Jul 27 2024 11:59PM. If you will need more time than this to complete your revisions, please reply to this message or contact the journal office at plosone@plos.org. Please include the following items when submitting your revised manuscript:A rebuttal letter that responds to each point raised by the academic editor and reviewer(s). You should upload this letter as a separate file labeled 'Response to Reviewers'.A marked-up copy of your manuscript that highlights changes made to the original version. You should upload this as a separate file labeled 'Revised Manuscript with Track Changes'.An unmarked version of your revised paper without tracked changes. You should upload this as a separate file labeled 'Manuscript'.

We look forward to receiving your revised manuscript.

Kind regards,

Elizabeth S. Mayne, M.D.

Academic Editor

PLOS ONE

Journal Requirements:

A clean copy of the edited manuscript (uploaded as the new *manuscript* file)”.

3. In the online submission form, you indicated that [All data analyzed and used during this study are available upon reasonable request]. 

Reviewers' comments:

Reviewer's Responses to Questions

**Comments to the Author**

1. Is the manuscript technically sound, and do the data support the conclusions?

Reviewer #1: No

Reviewer #2: Yes

2. Has the statistical analysis been performed appropriately and rigorously? 

Reviewer #1: No

Reviewer #2: Yes

3. Have the authors made all data underlying the findings in their manuscript fully available?

Reviewer #1: Yes

Reviewer #2: Yes

4. Is the manuscript presented in an intelligible fashion and written in standard English?

Reviewer #1: Yes

Reviewer #2: No

5. Review Comments to the Author

Reviewer #1: Abstract

Cochrane’s I2 is a secondary confirmation, it does not provide a p-value. Use tau-squared;

I2 should be stated as Cochrane’s I^2

Pooled proportion is 18.35 (95% CI: 14.89-21.81) calculated as follows:

18.35+1.96σ x (s/√n) = 18.35+1.96.(10.73/√37) =21.81 upper limit

18.35-1.96σ x (s/√n) = 18.35-1.96.(10.73/√37) =14.89 lower limit

s = sample standard deviation; n = 37

α = 0.025 x 37 = 0.925 ≈ 1.

This means that your funnel plot should not have more than 1 observation outside the left pseudo 95% limit and not more than 1 outside the right pseudo limit. You have 7 outside the left limit and 16 outside the right limit. This shows your distribution is highly non-normal, and that the parameters of a normal distribution do not apply. The funnel plot does not indicate that an Egger’s trim and fill has been done because the number of papers remains at 37, and the left half of the funnel is relatively empty. The funnel plot is thus unacceptable in all details.

You have used STATA 14.2 for your meta-analysis, which is an add-in using the meta command. This uses the DerSimonian and Laird approach (DL) for the pooled effects and the diamond. You have overall DL at p<0.000 and I^2 =97.1%. This does not show the detailed tests for heterogeneity such as Tau-squared, H-squared and Q (see Borenstein below). STATA 16 onwards incorporates meta-analysis in its suite. I would recommend you use the latest version (STATA 18). Read also the STATA manual on meta-analysis. Note that funnel plots are rarely successful (Borenstein et.al. (2009)) mainly because of incomplete spread of publications.

Consult the paper: Michael Borenstein (8 November, 2023) https://doi.org/10.1002/jrsm.1678 Avoiding common mistakes in meta-analysis: Understanding the distinct roles of Q, I-squared, tau-squared, and the prediction interval in reporting heterogeneity (Wiley)

Figure 2 Forest plot of pooled mortality. Although the random effects model applies, and publications therefore have similar weights, the effects confirm the funnel plot’s conclusion of a very non-normal distribution. The top 5 effect means are: Habtamu 80.48; Palme et.al. 37.93; Birhan et.al. 35.39; Gebreyes 29.48; Belayneh et.al. 25.7. The bottom 5 effect means are: Ifa 4.40; Seyoum et.al. 4.45; Alemu et.al. 5.63; Balcha et.al. 6.15; Tola et.al. 7.70. Using the upper limit equation above 18.35+1.96*(10.73/√37) =21.81 upper limit and 18.35-1.96*(10.73/√37) =14.89 lower limit. In the top and bottom studies, Habtamu and Ifa respectively, we have:

The Habtamu mean has a z=35.2 above the mean, that is: 18.35+35.2*(10.73306/37^0.5) ≈ 80.48

The Ifa mean has a z=7.9 below the mean, that is 18.35-7.9*(10.73306/37^0.5) ≈ 4.40.

Applying these to the other 8 studies highlights the problems of the outliers in the funnel plot.

The conclusion on this forest plot is that the 37 studies constitute a highly non-normal distribution and that the 18.35% prevalence is invalid as a measure for Ethiopia as a whole.

On the subgroup analysis, Tigray region has the highest mortality 31.86% (95% CI 8.83-54.88). The abstract does not give the region with the lowest mortality and Figure 4 forest plot of subgroup analysis by region is not readable nor is the click download for Figure 4 readable so I cannot comment on this. The CI spread of 8.83 – 54.88 for Tigray indicates that this region must have a mixture of very high and very low prevalences. The conclusion is that Ethiopia does not have a consistent monitoring, recording, treatment, and health system, and an overall 18.35% prevalence is misleading because of the extreme outliers and imbalance in the regions.

Hazard ratios: the abstract should indicate that hazard ratios have been used and the various ratios for predictors like HR, for example:

In the AHR region poor adherence to ART should indicate HR 2.50(95% CI: 1.28-4.91) also for the other HRs.

Hazard ratios are computed in the underlying papers and are the ratios between treatment curves (HIV/TB) and curves for corresponding curves for controls (non-HIV/TB). A problem is that the curves in the underlying papers have different time periods, and this may affect comparability. Hazard ratio curves are often linearized by using log HR or log RR. The various graphs for predictors use The DL method and show the overall DL heterogeneity measure but not the component measures (see DerSimonian and Laird approach (DL) comment above).

MeSH has been used to search and select papers on the basis of key words and phrases. Only 37 out of 886 studies qualified. There is no test that the key words selected satisfy the relationship between the dependent variable mortality and the independent variables like anaemia, poor ART adherence and lack of co-trimoxazole therapy. Independent variables such as low CD4 count, poor nutritional status, male sex, female sex worker, early initiation of ART for TB patients in various papers have not been included in this paper and have not been included in the MeSH search.

The same independent variables have been applied across categories but would vary between children and adults. A logistic regression should be done to ensure that the independent variables demarcate mortality and non-mortality and that these independent variables are included in the MeSH search and model.

The logistic regression has the form P=(e^(a +bx))/(1+e^(a+bx)); P takes on binary values

(1= mortality; 0= no-mortality) and b is the vector of independent variables that take continuous values.

Background gives WHO mortality rates, rates for Lesotho, Zambia, children, African region, and South East Asia. These would most likely have different predictors so they are not strictly comparable. For clarity of presentation they should be shown in a table.

Spelling and grammar: change ‘systemic’ to ‘systematic’, ‘Anemia’ to ‘anaemia’

Reviewer #2: • There are no page and line numbers used in the document- the authors did not follow PLOS instructions and makes it difficult to refer.

• According to The United States National Institutes of Health, the United Nations and its Children’s Fund (UNICEF), and the European Union pediatrics are defined as persons younger than 18 years. Which guidelines did the authors use to categorize pediatrics up to 15 years?

• Tabe 1: years must be corrected for ID 9 and 11.

• Reference the software used to analyse eg. Endnote version 9.

• The authors need to be consistency when referring to HIV-TB coinfection/TB-HIV co-infection/HIV/TB co-infection.

• P value should be written consistently throughout the manuscript with a symbol = or ≤.

• All the abbreviations must be written in full before abbreviation.

• Page 13: The authors referenced 6 studies but mentioned that five studies were included to examine the association between anemia and mortality.

• Page 14: Last sentence under “Poor adherence” should read “For this predictor publication bias was not performed due to small number of studies”.

• There are some spelling and grammatical errors on the manuscript that need to be corrected eg, page 14 under extrapulmonary tuberculosis: HIIV-TB should read HIV-TB.

• In the discussion, when comparing your studies with others, indicate the location of the systematic review and meta-analysis you are comparing too. Paragraph 2 and 6.

• Indicate when was the test-and treat strategy introduced globally and in Ethiopia?

• Discussion: The pecentages after the countries should be inside the brackets

Eg. This finding is in line with studies conducted in Sub-Saharan Africa (SSA) (18.1%) (1), Europe and Latin America (19%) (63), and systematic review and meta-analysis (15%) (64). However, this finding was lower than a study conducted in Eastern Europe (29%) (63), Brazil (27.4%) (65), and a systematic review and meta-analysis (33.56%)

• Discussion, paragraph 3 “This discrepancy may be related to limited access to appropriate therapy, …………...”- This statement is contrary to the above where the author mentioned that “The other possible justification is the study period, those studies were incorporated studies conducted before the commencement of the test-and-treat strategy, but in the present study, most of included studies were conducted after test and treat all strategy which is a proven strategy in reducing mortality.

6. PLOS authors have the option to publish the peer review history of their article (what does this mean?). If published, this will include your full peer review and any attached files.

Reviewer #1: **Yes: **Anthony Leland Hamilton Mayne

Reviewer #2: No

---

## [Author Response · Author response to Decision Letter 0]

10 Jul 2024

Authors' response to the Editor, and Reviewers' concern 

Title: Mortality and predictors among HIV/TB co-infected patients in Ethiopia: A Systematic Review and Meta-analysis.

Manuscript ID: PONE-D-24-06919

Date: July, 10, 2024.

Subject: Revision of the manuscript. 

Thank you very much dear editor and reviewers for your valuable and constructive suggestions and comments. We found that the comments are very helpful and constructive to further improve the manuscript. This is our point-by-point response of authors to the editor and reviewers’ suggestions and concerns about the manuscript entitled “Mortality and predictors among HIV/TB co-infected patients in Ethiopia: A Systematic Review and Meta-analysis” which has a manuscript ID of “PONE-D-24-06919” given by the journal. It is known that the manuscript has been reviewed by reviewers and sent back to the authors for revision and resubmission. As authors of this manuscript, the comments and concerns raised by the reviewers and editor are highly important enabling us to improve the quality and plausibility of the manuscript. To do so, we have addressed all of the reviewer's concerns point by point. Therefore, we are very much pleased to resubmit the revised version of the manuscript for further revision process and facilitation of its publication on PLOS ONE. 

We look forward to hear from you at your earliest convenience. 

With best regards.

Corresponding Author

Wubet Tazeb Wondie E-mail: wubettazeb27@gmail.com.

On behalf of Co-authors.

Academic Editor’s comments and Authors’ response.

Concern 1: When submitting your revision, we need you to address these additional requirements. Please ensure that your manuscript meets PLOS ONE's style requirements, including those for file naming.

Authors Response: Thank you dear editor for your concern. We have accepted and corrected in the revised manuscript. 

Concern 2: We suggest you thoroughly copyedit your manuscript for language usage, spelling, and grammar. If you do not know anyone who can help you do this, you may wish to consider employing a professional scientific editing service. 

Authors Response: Thank you dear editor for your concern. We have accepted and corrected in the revised manuscript as much as possible.

Concern 3: In the online submission form, you indicated that [All data analyzed and used during this study are available upon reasonable request]. 

Authors response: Thank you for your suggestion. We have accepted and the data is freely available as supplementary information. 

Concern 4: Please include captions for your Supporting Information files at the end of your manuscript, and update any in-text citations to match accordingly.

Authors response: Thank you dear editor for your suggestions. We have accepted and included cations of supporting information at the end of the manuscript.

Point-by-point Response Letter

Dear editorial office of PLOS ONE we have presented our point-by-point response in a way that the reviewers' concern is depicted first, and the authors' response has been given immediately next to it.

Reviewer 1 concerns and Authors’ response.

Concern 1: Abstract Cochrane’s I2 is a secondary confirmation, it does not provide a p-value. Use tau-squared. 

Authors response: Thank you dear reviewer for your golden suggestion. We have accepted and corrected using STATA 18 in the revised manuscript based on your recommendation. 

Concern 2: I2 should be stated as Cochrane’s I^2

Authors response: Thank you for your valuable suggestion. We have accepted and corrected through out the modified manuscript.

Concern 3: Pooled proportion is 18.35 (95% CI: 14.89-21.81) calculated as follows:

18.35+1.96σ x (s/√n) = 18.35+1.96.(10.73/√37) =21.81 upper limit

18.35-1.96σ x (s/√n) = 18.35-1.96.(10.73/√37) =14.89 lower limit

s = sample standard deviation; n = 37.

α = 0.025 x 37 = 0.925 ≈ 1.

This means that your funnel plot should not have more than 1 observation outside the left pseudo 95% limit and not more than 1 outside the right pseudo limit. You have 7 outside the left limit and 16 outside the right limit. This shows your distribution is highly non-normal, and that the parameters of a normal distribution do not apply. The funnel plot does not indicate that an Egger’s trim and fill has been done because the number of papers remains at 37, and the left half of the funnel is relatively empty. The funnel plot is thus unacceptable in all details.

Authors response: Thank you dear reviewer for your valuable and golden concern. As you have said in the random effects model using the Dersimonian Liard method, there are 7 studies out of the left limit and 16 studies out of the right limit, and the left side funnel plot is relatively empty, this suggests a high degree of heterogeneity. So, to assess the publication bias we have used the Eggers test for small study effect, and non-parametric trim and fill analysis. Accordingly, in the non-parametric trim and fill analysis 3 studies have been imputed and a total of 40 studies were included in the observed and imputed studies in the random effects Restricted Maximum Likelihood model. Due to the non-normal distribution of the data we have used non-parametric trim and fill analysis rather than the standard trim and fill analysis. In addition, we used a Galbraith plot to identify the presence of heterogeneity. 

Concern 3: You have used STATA 14.2 for your meta-analysis, which is an add-in using the meta command. This uses the DerSimonian and Laird approach (DL) for the pooled effects and the diamond. You have overall DL at p<0.000 and I^2 =97.1%. This does not show the detailed tests for heterogeneity such as Tau-squared, H-squared and Q (see Borenstein below). STATA 16 onwards incorporates meta-analysis in its suite. I would recommend you use the latest version (STATA 18). Read also the STATA manual on meta-analysis. Note that funnel plots are rarely successful (Borenstein et.al. (2009)) mainly because of incomplete spread of publications.

Consult the paper: Michael Borenstein (8 November, 2023) https://doi.org/10.1002/jrsm.1678 Avoiding common mistakes in meta-analysis: Understanding the distinct roles of Q, I-squared, tau-squared, and the prediction interval in reporting heterogeneity (Wiley).

Authors response: Thank you dear reviewer for your golden suggestions. We have accepted and used the latest version STATA18 based on your recommendation, and we tried to show Tau-squared, H-squared, and Q heterogeneity tests

Concern 4: Figure 2 Forest plot of pooled mortality. Although the random effects model applies, and publications therefore have similar weights, the effects confirm the funnel plot’s conclusion of a very non-normal distribution. The top 5 effect means are: Habtamu 80.48; Palme et.al. 37.93; Birhan et.al. 35.39; Gebreyes 29.48; Belayneh et.al. 25.7. The bottom 5 effect means are: Ifa 4.40; Seyoum et.al. 4.45; Alemu et.al. 5.63; Balcha et.al. 6.15; Tola et.al. 7.70. Using the upper limit equation above 18.35+1.96*(10.73/√37) =21.81 upper limit and 18.35-1.96*(10.73/√37) =14.89 lower limit. In the top and bottom studies, Habtamu and Ifa respectively, we have:

The Habtamu mean has a z=35.2 above the mean, that is: 18.35+35.2*(10.73306/37^0.5) ≈ 80.48

The Ifa mean has a z=7.9 below the mean, that is 18.35-7.9*(10.73306/37^0.5) ≈ 4.40.

Applying these to the other 8 studies highlights the problems of the outliers in the funnel plot.

The conclusion on this forest plot is that the 37 studies constitute a highly non-normal distribution and that the 18.35% prevalence is invalid as a measure for Ethiopia as a whole.

Authors response: Thank you dear reviewer for your valuable concern. As you have said there are outliers Habtamu et, al, Ifa … The main reason for this outlier and variation is in the Tigray region there is poor adherence and awareness of ART which increases the mortality of HIV-infected patients particularly in co-infected patients that means the Habtamu’s study is higher than the other. On the other hand, in SNNPR, the community has a good awareness about ART and the importance of good adherence, which reduces the mortality rate, that means the Ifa et al study (a study conducted in SNNP), is lower than the other. 

By considering the variation of those studies, we have used a random effect restricted maximum likelihood model. To identify the potential source of heterogeneity, a sensitivity analysis by excluding the outliers to assess the impact on the pooled estimate was conducted, and in the sensitivity analysis, all of the leave-one-out estimates were within the confidence interval of the pooled estimate. In addition, a sub-group analysis to explore the source of heterogeneity & outliers was done. If we exclude those studies, the overall pooled mortality could be compromised. On the other hand, it could not be representative of the mortality rate of HIV-TB co-infected patients in Ethiopia. So, by considering this, we include those studies found in the country and we did the aforementioned analysis.

Concern 5: On the subgroup analysis, Tigray region has the highest mortality 31.86% (95% CI 8.83-54.88). The abstract does not give the region with the lowest mortality and Figure 4 forest plot of subgroup analysis by region is not readable nor is the click download for Figure 4 readable so I cannot comment on this. The CI spread of 8.83 – 54.88 for Tigray indicates that this region must have a mixture of very high and very low prevalences. The conclusion is that Ethiopia does not have a consistent monitoring, recording, treatment, and health system, and an overall 18.35% prevalence is misleading because of the extreme outliers and imbalance in the regions.

Authors’ response: Thank you for your concern. We have accepted and corrected the revised manuscript and tried to show a visible forest plot. As you have said there are outliers, particularly in Tigray region, and Ethiopia as low-income countries don’t have consistent monitoring and health systems for HIV-TB co-infected patients. Our study tried to show this finding (problem) for the concerned body in the country. So, to tackle this public health problem we accept this finding with its limitations. 

Concern 6: Hazard ratios: the abstract should indicate that hazard ratios have been used and the various ratios for predictors like HR, for example: In the AHR region poor adherence to ART should indicate HR 2.50(95% CI: 1.28-4.91) also for the other HRs.

Authors response: Thank you dear reviewer for your suggestions. We have accepted and corrected the revised manuscript. 

Concern 7: Hazard ratios are computed in the underlying papers and are the ratios between treatment curves (HIV/TB) and curves for corresponding curves for controls (non-HIV/TB). A problem is that the curves in the underlying papers have different time periods, and this may affect comparability. Hazard ratio curves are often linearized by using log HR or log RR. The various graphs for predictors use The DL method and show the overall DL heterogeneity measure but not the component measures (see DerSimonian and Laird approach (DL) comment above).

Authors response: Dear reviewer thank you for your valuable concern. As you have said the difference in time period in the underlying paper affects the comparability. To enhance comparability and reduce bias we have linearized the hazard ratio using the natural logarithm of hazard ratio (HR) then we calculate the standard error of log hazard. We use the log hazard and the standard error for meta-analysis. Finally, the pooled log hazard ratio was back-transformed to the original scale by taking the exponent of the pooled log Hazard ratio. 

Concern 8: MeSH has been used to search and select papers on the basis of key words and phrases. Only 37 out of 886 studies qualified. There is no test that the key words selected satisfy the relationship between the dependent variable mortality and the independent variables like anaemia, poor ART adherence and lack of co-trimoxazole therapy. Independent variables such as low CD4 count, poor nutritional status, male sex, female sex worker, early initiation of ART for TB patients in various papers have not been included in this paper and have not been included in the MeSH search.

Authors response: Thank you dear reviewer for your concern. Our objective is mortality and predictors among HIV-TB co-infected patients. Variables like anemia, poor adherence, lack of co-trimoxazole therapy, low CD4 count, poor nutritional status, male sex, female sex worker, and early initiation of ART……… are all predictors, and those primary studies reported these variables as predictors. When we use the key terms predictor, associated factor, risk factor, and determinant in our search it refers to those variables. We tried to use MeSH terms for the main keywords but, we didn’t get the MeSH term except the first key term. We have used comprehensive searching by combining the Free text words with the MeSH term and related terms. After searching those studies, we extracted all predictors and did the analysis. Finally, out of 20 predictors, four predictors were significantly associated with the outcome variables (Mortality). We have stated it in the manuscript. To not miss studies that report the mortality and predictors among HIV-TB co-infected patients, we have used different keywords as shown in the supplementary file. 

Concern 9: The same independent variables have been applied across categories but would vary between children and adults. A logistic regression should be done to ensure that the independent variables demarcate mortality and non-mortality and that these independent variables are included in the MeSH search and model.

The logistic regression has the form P= (e^(a +bx))/(1+e^(a+bx)); P takes on binary values (1= mortality; 0= no-mortality) and b is the vector of independent variables that take continuous values.

Authors response: Thank you for your concern. The primary studies in children and adult have similar categories, due to this we applied the same independent variables. The primary studies that report Predictors are cohorts either retrospective or prospective (But studies that report the proportion of mortality are either cross-sectional or cohort). So, the primary studies are categorizing independent variables into mortality and non-mortality and they use survival analysis. Therefore, in my view, logistic regression is not necessary. In addition, by considering the difference between Pediatrics and Adult we were trying to stratify the analysis of children and adult groups. However, the main problem we face is that almost all of the predictors in either of the groups have either one or two studies that report the association between the required predictor and the outcome variables, and analyzing one or two studies is impossible.

Concern 10: Background gives WHO mortality rates, rates for Lesotho, Zambia, children, African region, and South East Asia. These would most likely have different predictors so they are not strictly comparable. 

Authors’ response: Thank you for your concern. But all of these ideas were narrated to show the magnitude and severity of the problem from global to local context in children and adults because our study includes all population groups.

Concern 11: For clarity of presentation they should be shown in a table. 

Authors response: Thank you dear reviewer for your golden suggestion. We tried to present the invisible forest plot by tables.

Concern 12: Spelling and grammar: change ‘systemic’ to ‘systematic’, ‘Anemia’ to ‘anaemia’

Authors response: Thank you dear review for your concern. We have accepted and corrected the revised manuscript.

N.B: In this revised manuscript we analyzed the data using STATA 18 and we used the random effects model with the Restricted Maximum Likelihood (REML) method, rather than the previous Dersimonian Liard method. Due to this, there are some decimal point changes in the revised manuscript.

Dear reviewer, thank you for your golden concerns. All of yo

---

## [Decision Letter · Decision Letter 1]

3 Sep 2024

PONE-D-24-06919R1Mortality and predictors among HIV-TB co-infected patients in Ethiopia: A Systematic Review and Meta-analysis.PLOS ONE

Dear Dr. Wondie,

Thank you for submitting your manuscript to PLOS ONE. After careful consideration, we feel that it has merit but does not fully meet PLOS ONE’s publication criteria as it currently stands. Therefore, we invite you to submit a revised version of the manuscript that addresses the points raised during the review process.

**The statistical reviewer still identifies significant concerns with the analysis here particularly, for example, with the construction of the forest plots. Since this is a systematic review, these plots are essential to the underlying analysis.**

We look forward to receiving your revised manuscript.

Kind regards,

Elizabeth S. Mayne, M.D.

Academic Editor

PLOS ONE

Reviewers' comments:

Reviewer's Responses to Questions

**Comments to the Author**

1. If the authors have adequately addressed your comments raised in a previous round of review and you feel that this manuscript is now acceptable for publication, you may indicate that here to bypass the “Comments to the Author” section, enter your conflict of interest statement in the “Confidential to Editor” section, and submit your "Accept" recommendation.

Reviewer #1: (No Response)

Reviewer #2: All comments have been addressed

2. Is the manuscript technically sound, and do the data support the conclusions?

Reviewer #1: No

Reviewer #2: Yes

3. Has the statistical analysis been performed appropriately and rigorously? 

Reviewer #1: No

Reviewer #2: Yes

4. Have the authors made all data underlying the findings in their manuscript fully available?

Reviewer #1: Yes

Reviewer #2: Yes

5. Is the manuscript presented in an intelligible fashion and written in standard English?

Reviewer #1: No

Reviewer #2: Yes

6. Review Comments to the Author

**Reviewer #1: **PLoS One Mortality and predictors among HIV-TB co-infected patients in Ethiopia: A Systematic Review and Meta-analysis

Pone-D-24-06919R1

Lines

31 systematic.

38 Cochrane Q and I2 test statistics.

42 A total of 886 studies were identified, using database searches and keywords. Of these, 37 met the criteria for inclusion. (omit “and were included in this study”).

44-49 The APA style is 95% CI [4.32, 9.26]. Note that the STATA 18 forest plot style is 4.45[3.24, 5.66] effect size with 95% CI, which is also acceptable.

44-49 The APA style for hazard ratios is (HR=2.02, 95% CI [1.7, 2.5]).

44-49 APA style for p-values is p<.001 or p≤.05 not p=.00 i.e. state a limit. I know STATA 18 gives the form p=0.00 in the forest plots, but in a formal report, use the APA style (p=0.00 becomes p<.01).

50 Nearly one-fifth? How much? present the value, for example, 19.6%.

The introduction needs to be rewritten, please refer this to an experienced editor. What is the African region (line 69) and South-East Asian region, line (69). Line 70, what is WHO Africa region? These are vague.

73 high-HIV-TB burden countries like Ethiopia. This is vague.

74 $78, is this US dollars?

75 Several studies names 7 factors. Are these the only ones? Did all the studies deal with the same 7 factors? (what about NCDs, cancers?). (No in fact 20 were considered in line 265 – you need to disclose this fact here).

85 46% were getting ART. Does this apply to Ethiopia?

89 Different corners? State "Ethiopia"

96 Systematic.

123 Capitalisation of headings is inconsistent, for example, Eligibility Criteria, capitalises both words. Elsewhere only the first is capitalised.

124 Systematic.

127 Too many hyphens.

128 English (drop language).

130 P: HIV-TB etc.

143 Software eliminated duplication of studies.

144 Briggs (JBI) critical appraisal tools.

153 the pooled hazard ratios (PHR) with 95% confidence intervals…

158 A restricted maximum likelihood (REML) means a restricted conclusion. How does this affect your study?

159 Please read the Cochrane handbook 10.4.3.1 at Cochrane.org https://handbook-5-

1.cochrane.org/chapter_10/10_4_3_1_recommendations_on_testing_for_funnel_plot_asymmetry.htm Your details are

not clear and application of tests may not be valid. Tests for heterogeneity are only approximate too.

166 p≤.05 APA style.

168 PHR already defined in line 153.

194 Table 1. Style is inconsistent. Mortality % column studies should all use the same number of decimal points.

194 ID 6 remove “all over”. Stating “Ethiopia” is sufficient.

194 Some authors have initials, others do not.

194 ID 11 what is 2012S?

194 Line up follow-up period dates.

194 IDs 14, 17, 18, 19, 20, 22, 23, 24 … 37 have commas after name, others do not

199 (S3). ( ?

200 Pooled proportion of mortality

202 Death rates of 4.4% …. to 80.45%.

202 to 205 A forest plot weights studies by relative sample size, which is not the same thing as prevalence, which is weighted on relative population size. If the 80.45% applies to 90% of the population of Ethiopia it has a different effect than if it applied to (say) only 10% of the overall population. In Table 3 on page 13, the 18.42% mortality for Ethiopia is weighted on sample sizes of included studies. Table 3 shows number of studies, event, and sample sizes. The discussion of Table 3 should disclose how weighting was applied, for example, did weighting include information in all three columns, or was it based, as I suspect, on sample size only. The 18.42% must be described as weighted on sample sizes (if this is the case), and that this is used as a best proxy for prevalence. Prevalence takes relative population sizes of regions into account. If you do not have these figures, you must qualify your statistic, otherwise your study is invalid.

(203 I know the Tigray region was weighted in the forest plot to get a weighted overall effect size and 95% CI. How have you done your weighting of Tigray and other regions to get to your 18.42% overall mortality for Ethiopia? (See comment above).

205-208 For p-value and CI disclosure, see APA style in lines 44-49.

213-216 See APA style for disclosure.

214 Fig 3 shows many studies outside the 95% CI lines, so the trim and fill exercise is unsuccessful. You need to discuss this or acknowledge the problem and state how you have overcome this. Your conclusion on the 18.42% must be qualified as having been based on a heterogeneous spread of papers. It may help to justify the 18.42% by basing it on population sizes and minimising the effect of the Tigray region’s outlier effect by stating, for example, that Tigray forms only 5% of the Ethiopian population. How have other countries arrived at their prevalence rates?

216 (Table 2). ( ?

219 Table 2 does not state a conclusion and is meaningless in terms of the problem in line 214.

223-226 See APA style for disclosure (see lines 44-49).

227 The sensitivity analysis to resolve the outliers in the Galbraith plot is unsuccessful because many studies lie outside the 95% CI bands. You need to explain why this is the case, if this is important, and how you have discounted this in your conclusion.

233 to 238 See APA disclosure of p-values and confidence intervals.

242-243 See the forest plot presentation of effect size and 95% CI used by STATA 18.

242-243 The 18.42[14.27, 22.57] for Ethiopia is based on sample size. Prevalence needs to be weighted by relative population size (see discussions above).

248 Remove furthermore.

250 Remove accordingly.

251-254 For effect size and CI, use STATA 18 forest plot form, for formal disclosure p-value use the APA form.

265 20 predictors assessed. These are not mentioned in your abstract on page 2. Only the 7 significant ones are shown in lines 50-53. The abstract on page 2 is incomplete. You are concluding on something that has not been included in your opening statements of aims.

265-272 You cannot use a forest plot to assess which predictors are significant. A forest plot is a meta-analysis of a particular property in a set of publications. It is not a direct statistical model to assess individual significance in a set of multiple variables which are likely to be interdependent. This can only be done by regression models, ANOVA models or other multivariable statistical models. You need to design a matrix of dependent and independent variables and a regression analysis (for example a multiple linear regression, or preferably a logistic regression). You are attempting to gauge significance of individual variables by subgroup forest plots. The more subgroup analyses you do, the smaller is the sample size of data which you are testing, which may affect significance achieved, if sample sizes are too small. The testing of 20 variables as independent predictors is incorrect unless you can demonstrate they are independent. A correct test is to include these into a matrix with a column for the dependent variable (HIV1-TB) and to determine model fit (root mean square values), F statistics and p values and to remove variables with p values >.05

265-272 You cannot assess predictors singly. There will be some collinearity (dependence between one or more predictors) See comment above.

325-336 comparisons between countries. You need to justify why these are comparable. Were they done on the same basis as your analysis? Are they claiming prevalence or are they based on your sample weighting basis?

337 heterogeneity indicates to what extent studies do not come from a common source or population. The Tigray region is an outlier, and there are relatively few studies to do a test of more than a few predictors. This is a major issue, and a hazard ratio may not be significant because of this.

337 If there is interdependence between predictors, it may affect the significance of the hazard ratios you are computing as isolated predictors.

387 Strengths and limitations. This indicates major limitations, for example, high heterogeneity, unavailability of studies and inconsistent measurements. This means that the result of 18.42% proportion of mortality is in doubt and that your study has failed. This is a significant problem. You can circumvent this problem by qualifying your figure as based on a best proxy and justifying it by referring to other countries who have used the same basis.

**Reviewer #2:** The authors addressed all my comments to my satisfaction and the manuscripts is written in a scientific way.

7. PLOS authors have the option to publish the peer review history of their article (what does this mean?). If published, this will include your full peer review and any attached files.

Reviewer #1: **Yes: **Anthony L.H. Mayne

Reviewer #2: No

---

## [Author Response · Author response to Decision Letter 1]

26 Sep 2024

Authors' response to the Editor, and Reviewers' concern 

Title: Mortality and predictors among HIV/TB co-infected patients in Ethiopia: A Systematic Review and Meta-analysis.

Manuscript ID: PONE-D-24-06919.

To: PLOS ONE JOURNAL

Date: September 26, 2024.

Subject: Revision of the manuscript. 

Thank you, very much dear editor and reviewers, for your valuable and constructive suggestions and comments. We found that the comments are very helpful and constructive to further improve the manuscript. This is our point-by-point response of authors to the editor and reviewers’ suggestions and concerns about the manuscript entitled “Mortality and predictors among HIV/TB co-infected patients in Ethiopia: A Systematic Review and Meta-analysis” which has a manuscript ID of “PONE-D-24-06919” given by the journal. It is known that the manuscript has been reviewed by reviewers and sent back to the authors for revision and resubmission. As authors of this manuscript, the comments and concerns raised by the reviewers and editor are highly important enabling us to improve the quality and plausibility of the manuscript. To do so, we have addressed all of the reviewer's concerns point by point. Therefore, we are very pleased to resubmit the revised version of the manuscript for further revision process and facilitation of its publication on PLOS ONE. 

We look forward to hearing from you at your earliest convenience. 

With best regards.

Corresponding Author

Wubet Tazeb Wondie E-mail: wubettazeb27@gmail.com.

On behalf of Co-authors.

Point-by-point Response Letter

Dear editorial office of PLOS ONE we have presented our point-by-point response in a way that the reviewers' concern is depicted first, and the authors' response has been given immediately next to it.

Reviewer 1 concerns and Authors’ response.

Concern 1: Line 31 systematic.

Authors response: Dear esteemed reviewer, we thank you for your valuable suggestion. We have accepted and corrected the revised manuscript. 

Concern 2: Line Cochrane Q and I2 test statistics.

Authors response: Dear reviewer, thank you for your suggestion. We have accepted and corrected the revised manuscript. 

Concern 3: Line 42 A total of 886 studies were identified, using database searches and keywords. Of these, 37 met the criteria for inclusion. (omit “and were included in this study”).

Authors response: Dear reviewer, thank you for your suggestion. We have accepted and corrected the revised manuscript. 

Concern 4: 44-49 The APA style for hazard ratios is (HR=2.02, 95% CI [1.7, 2.5])

Authors response: Dear reviewer, we thank you for your valuable suggestion. We have accepted and corrected the revised manuscript. 

Concern 5: Line 44-49 APA style for p-values is p<.001 or p≤.05 not p=.00 i.e. state a limit. I know STATA 18 gives the form p=0.00 in the forest plots, but in a formal report, use the APA style (p=0.00 becomes p<.01).

Authors response: Dear esteemed reviewer, thanks for your concern. We have accepted and corrected the revised manuscript. 

Concern 6: Line 50 Nearly one-fifth? How much? present the value, for example, 19.6%.

 Author response: Dear reviewer. We thank you for your valuable suggestion. In response to your suggestion, we have accepted and corrected the revised manuscript.

Concern 7: The introduction needs to be rewritten, please refer this to an experienced editor. What is the African region (line 69) and South-East Asian region, line (69). Line 70, what is WHO Africa region? These are vague.

Authors response: Dear esteemed reviewer, we thank you for your valuable concern and suggestion. We have accepted and rewritten the introduction. We kindly request you to review the revised manuscript.

Concern 8: Line 73 high-HIV-TB burden countries like Ethiopia. This is vague.

Authors response: Dear esteemed reviewer, we thank you for your important concern. We have accepted and corrected our write-up.

Concern 9: Line 74 $78, is this US dollars?

Authors response: Dear reviewer, for your concern. Yes, it is based on US dollars. 

Concern 10: Line 75 Several studies names 7 factors. Are these the only ones? Did all the studies deal with the same 7 factors? (what about NCDs, cancers?). (No in fact 20 were considered in line 265 – you need to disclose this fact here).

Authors response: Dear esteemed reviewer, we thank you for your valuable suggestion. We have accepted your suggestion and added the rest predictors in the revised manuscript. 

Concern 11: Line 85 46% were getting ART. Does this apply to Ethiopia?

Authors response: Dear reviewer, we thank you for your valuable concern. This figure is a global figure and it may not fully represent Ethiopia so we removed it. 

Concern 12: Line 89 Different corners? State "Ethiopia

Authors response: Thank you dear esteemed reviewer for your concern. We have accepted and corrected in revised version of the manuscript.

Concern 13: Line 96 Systematic.

Author response: Dear thank you for your concern. We have accepted and corrected it.

Concern 14: Line 123 Capitalisation of headings is inconsistent, for example, Eligibility Criteria, capitalises both words. Elsewhere only the first is capitalised.

Authors response: Dear reviewer, we thank you for your suggestion for clarity of our manuscript. We have accepted your suggestions and corrected in the revised manuscript. 

Concern 15: Line 124 Systematic

Authors' Response: We have accepted and corrected in the revised manuscript.

Concern 16: Line 127 Too many hyphens.

Author response: Dear reviewer, thank you for your suggestion we have accepted and made corrections in the revised manuscript. 

Concern 17: Line 128 English (drop language).

Authors response: Dear honored reviewer, thank you for your valuable suggestion. We have accepted and corrected the revised manuscript.

Concern 18: Line 130 P: HIV-TB etc.

Authors response: Dear reviewer, we thank you for your suggestions. We have accepted your suggestion and corrected it in the revised manuscript.

Concern 19: Line 143 Software eliminated duplication of studies.

Authors response: Dear honored reviewer, we thank you for your suggestion. We have accepted and corrected it in the revised manuscript. 

Concern 20: Line 144 Briggs (JBI) critical appraisal tools.

Authors response: Dear esteemed reviewer, thank you for your suggestion. We have accepted and corrected it in the revised manuscript. 

Concern 21: Line 153 the pooled hazard ratios (PHR) with 95% confidence intervals.

Authors response: Dear reviewer, we thank you for your suggestion. We have accepted your suggestion and corrected it in the revised manuscript. 

CONCERN 22: Line 158 A restricted maximum likelihood (REML) means a restricted conclusion. How does this affect your study?

Authors Response: Dear esteemed reviewer, we express our gratitude for your concrete and golden concern. The most widely used method in the random effects model is the Der Simonian-Liard method. However, when the effect sizes are not normally distributed the restricted maximum likelihood (REML) method is more robust in providing estimates that are less influenced by outliers. A restricted maximum likelihood (REML) method is particularly effective for estimating study variance (heterogeneity) because it focuses on the residual after accounting for the fixed effects, which leads to better estimates. On the other hand, the Der Simonian-Liard method assumes the effect sizes are normally distributed, which can lead to biased estimates when the assumption is violated. 

In General, due to the non-normal distribution of our data, we prefer the restricted maximum likelihood (REML), to account for the heterogeneity and get a reliable estimate. The REML doesn’t give a restricted conclusion rather it gives an unbiased estimate in case of non-normal distribution. 

Concern 23: Line 166 p≤.05 APA style.

Authors response: Dear reviewer, we thank you for your suggestion. We have accepted and corrected it in the revised manuscript.

Concern 24: Line 168 PHR already defined in line 153.

Author response: Dear reviewer, we thank you for your suggestion. We have accepted your suggestions and deleted the repeated sentence. 

Concern 26: 194 Table 1. Style is inconsistent. Mortality % column studies should all use the same number of decimal points.

Authors response: Dear reviewer, we thank you for your valuable concern for the clarity of our study. We have accepted your suggestion and corrected it in the revised manuscript.

Concern 27: Line 194 ID 6 remove “all over”. Stating “Ethiopia” is sufficient.

Authors response: Dear reviewer, thank you for your suggestion. We have corrected it in the revised manuscript.

Concern 28: Lines 194 Some authors have initials, others do not.

Authors response: Dear reviewer, we thank you for your suggestions. We have accepted and corrected it. 

Concern 29: Line 194 ID 11 What is 2012S?

Authors response: Dear respected reviewer, we thank you for your suggestion. We have accepted and corrected it in the revised manuscript.

Concern 30: Line 194 Line up follow-up period dates.

Authors response: Thank you dear esteemed reviewer for your suggestions. We have accepted and corrected it in the revised manuscript. 

Concern 31: Line 194 IDs 14, 17, 18, 19, 20, 22, 23, 24 … 37 have commas after name, others do not.

Authors response: Dear respected reviewer, we thank you for your suggestion to the clarity of our manuscript. We have accepted your suggestion and corrected it in the revised manuscript.

Concern 32: (S3). ( ?

Authors response: Dear reviewer, we thank you for your suggestions. We have accepted and corrected it in the revised manuscript. 

Concern 33: Line 200 Pooled proportion of mortality

Authors response: Dear reviewer, thank you for your comment. We have accepted your comment and corrected it in the revised manuscript.

Concern 34: 202 Death rates the of 4.4% …. to 80.45%.

Authors response: Thank you dear esteemed reviewer for your suggestion. We have accepted and corrected it in the revised manuscript. 

Concern 35: Lines 202 to 205 A forest plot weights studies by relative sample size, which is not the same thing as prevalence, which is weighted on relative population size. If the 80.45% applies to 90% of the population of Ethiopia it has a different effect than if it applied to (say) only 10% of the overall population.

Authors Response: Dear esteemed reviewer, thank you for raising this golden concern. As you have said forest plot weight studies by sample size, inverse variance method, quality of studies……... rather than population size. During analysis, we consider these points, and in the present study, studies were weighted by their sample size. In this regard, we are ready to accept any of your recommendation’s dear reviewer. 

Concern 36: In Table 3 on page 13, the 18.42% mortality for Ethiopia is weighted on sample sizes of included studies. Table 3 shows number of studies, event, and sample sizes. The discussion of Table 3 should disclose how weighting was applied, for example, did weighting include information in all three columns, or was it based, as I suspect, on sample size only. The 18.42% must be described as weighted on sample sizes (if this is the case), and that this is used as a best proxy for prevalence. Prevalence takes relative population sizes of regions into account. If you do not have these figures, you must qualify your statistic, otherwise your study is invalid.

Authors Response: Dear reviewer, we thank you for your golden suggestion. We apologize for any confusion caused by our presentation. The weighting in Table 3 is based on the sample size. Just We showed this table to depict the proportion of mortality in each region in addition to the forest plot (Figure 5). We think that it creates confusion for the reader. So, we have removed those columns and presented the rest statistics. We hope the figure tells more. We are ready to accept any of your comments in this regard.

Concern 37: (203 I know the Tigray region was weighted in the forest plot to get a weighted overall effect size and 95% CI. How have you done your weighting of Tigray and other regions to get to your 18.42% overall mortality for Ethiopia? (See comment above).

Authors response: Dear reviewer, thank you for your valuable concern. To weight studies in the Tigray region, we use equal weights because there is a similar sample size to the other regions. We are ready to accept any of your suggestions in this regard. 

Concern 38: Lines 205-208 For p-value and CI disclosure, see APA style in lines 44-49.

Authors response: Thank you dear esteemed reviewer for your suggestions. We have accepted and corrected it in the revised manuscript.

Concern 39: Lines 213-216 See APA style for disclosure.

Authors response: Dear reviewer, thank you for your valuable suggestion for the clarity of our manuscript. We have accepted and corrected the revised manuscript.

Concern 40: Lines 214 Fig 3 shows many studies outside the 95% CI lines, so the trim and fill exercise is unsuccessful. You need to discuss this or acknowledge the problem and state how you have overcome this. Your conclusion on the 18.42% must be qualified as having been based on a heterogeneous spread of papers. It may help to justify the 18.42% by basing it on population sizes and minimizing the effect of the Tigray region’s outlier effect by stating, for example, that Tigray forms only 5% of the Ethiopian population. How have other countries arrived at their prevalence rates?

Authors Response: Thank you, dear esteemed reviewer, for your golden concern and critical view. As you have said in figure3, some studies were outside the 95% CI, because trim and fill and fill analysis is unsuccessful in case of substantial heterogeneity. Due to this, we acknowledge the presence of substantial heterogeneity as a limitation, and the pooled 18.42% mortality is based on heterogeneity spread. In this regard, we are ready to accept any of your potential recommendations. 

Concern 41: Line 216 (Table 2). ( ?

Authors' Response: Dear reviewer, thank you for your suggestions. We have accepted and corrected it in the revised manuscript. 

Concern 42: Line 219 Table 2 does not state a conclusion and is meaningless in terms of the problem in line 214.

Authors Response: Dear reviewer, we sincerely appreciate your critical concern. As you have said it doesn’t support the conclusion. Just we presented it to show the presence of publication bias and the trim and fill analysis. After considering this, it might create confusion for the readers so we have removed the table and the figure has been put as the previous one.

Concern 43: Line 227 The sensitivity analysis to resolve the outliers in the Galbraith plot is unsuccessful because many studies lie outside the 95% CI bands. You need to explain why this is the case, if this is important, and how you have discounted this in your conclusion.

Authors response: Dear reviewer, thank you for your critical concern. In the sensitivity analysis, the single estimate in each omitted study lies within the overall 95% CI (95% CI;14.27-22.57). None of the single estimates (effect size lies) in the omitted study lies out the 95% CI. Take a look at the single estimate of the omitted study in the sensitivity analysis and the overall estimate in the forest plot. 

Concern 44: Lines 233 to 238 See APA disclosure of p-values and confidence intervals.

Author response: Dear, we thank you for your concern. We have accepted and corrected in the revised manuscript.

Concern 45: Lines 242-243 See the forest plot presentation of effect size and 95% CI used by STATA 18.

Authors response: Dear thank you for your concern. We have seen it. To some extent, it is not visible. We have shown a table to support this figure (Table 3). 

Concern 46: Line 242-243 The 18.42[14.27, 22.57] for Ethiopia is based on sample size. Prevalence needs to be weighted by relative population size (see discussions above).

Authors response: Dear reviewer, we sincerely appreciate your critical comment. As we have mentioned above, in our study weighting is based on sample size. We hope the forest plots will tell more. 

Concern 47: Lines 248 Remove furthermore.

Authors response: Dear esteemed reviewer, thank you for your suggestions. We have accepted and corrected it. 

---

## [Decision Letter · Decision Letter 2]

29 Oct 2024

PONE-D-24-06919R2Mortality and predictors among HIV-TB co-infected patients in Ethiopia: A Systematic Review and Meta-analysis.PLOS ONE

Dear Dr. Wondie,

Thank you for submitting your manuscript to PLOS ONE. After careful consideration, we feel that it has merit but does not fully meet PLOS ONE’s publication criteria as it currently stands. Therefore, we invite you to submit a revised version of the manuscript that addresses the points raised during the review process.

 This manuscript was again sent for specialised statistical review and the statistician has made several comments which still need to be corrected. These include consistent reporting of results, the correct grouping (within and between groups) and some of the extrapolations of the data.

We look forward to receiving your revised manuscript.

Kind regards,

Elizabeth S. Mayne, M.D.

Academic Editor

PLOS ONE

Reviewers' comments:

Reviewer's Responses to Questions

**Comments to the Author**

1. If the authors have adequately addressed your comments raised in a previous round of review and you feel that this manuscript is now acceptable for publication, you may indicate that here to bypass the “Comments to the Author” section, enter your conflict of interest statement in the “Confidential to Editor” section, and submit your "Accept" recommendation.

Reviewer #1: (No Response)

2. Is the manuscript technically sound, and do the data support the conclusions?

Reviewer #1: Partly

3. Has the statistical analysis been performed appropriately and rigorously? 

Reviewer #1: No

4. Have the authors made all data underlying the findings in their manuscript fully available?

Reviewer #1: Yes

5. Is the manuscript presented in an intelligible fashion and written in standard English?

Reviewer #1: No

6. Review Comments to the Author

Reviewer #1: Lines

General: reporting of data especially p-values and 95% CLs is inconsistent. STATA shows exact p-values, p=0.00 so references to forest plots must show the exact values. Sometimes you adopt the STATA representation of CIs as in the square brackets in line 246 and sometimes your depiction with round brackets. A p-value gives the probability that a statistic like i^2 is a chance value. If (say) i^2=98.002, which indicates high heterogeneity, p=0.11 is not significant, because it is higher than p=0.05 and we cannot conclude that the i^2 is significant or not (a p-value is usually computed by a chi-square distribution, and a statistic such as i2 does not determine its own p-value).

45-46 the lowest pooled mortality was reported for two general studies in Ethiopia 11.95 (95% CI; 4.19-19.00).

51 … 18.42% (95% CI; 14.27-22.57), i2=98.02%. The lowest individual study mortality was 4.40 (95%CI; 1.21-7.50) and the highest, 80.48 (95%CI; 75.12-85.84), which indicated significant interstudy heterogeneity. This was confirmed by the funnel and Galbraith plots, where 22 out of the 37 studies lay outside the central pseudo-95% region. The tau-squared value of 160.41 indicated there was significant within study variability. In the subgroup analysis…

65 …which accounts for 30%...

164-167 (this is a conceptual section, not a results section). …to assess the presence of a small study effect, an Egger’s test was done. The trim and fill funnel plot analysis was confirmed by a Galbraith plot, and the studies lying inside and outside the central 95% pseudo-CI region was agreed to the distributional features in the pooled mortality forest plot in Fig 2.

205-207 rewrite this (note that p values in forest plots are exact in STATA forest plots).

205-207 significant heterogeneity was observed between studies with a Q=123.16; and i^2 =98.02 and H^2 =50.46; P=0.00. Tau-squared=160.41 indicated significant heterogeneity within studies.

211-226 I suggest a rewrite of these lines as follows -

211-215 An Egger’s test value, P<0.001, indicated a significant publication bias, and highly non-normal distribution. A funnel plot confirmed both the bias and the non-normal distribution, but the non-parametric trim and fill analysis failed to resolve the bias because 22 of the 37 studies and the 3 imputed studies remained outside the pseudo 95% CIs. Standard errors to effect size (mortality) varied between 6.5 standard errors and 0.5 standard errors before and after the three imputed studies. A Galbraith plot confirmed the results of the funnel plot.

216 Fig 3: Non-parametric funnel plot with fill and trim analysis

217 Fig 4: Galbraith plot

218-226 Handling heterogeneity

Because of the high heterogeneity between and within the 37 studies, a forest plot was prepared using the REML method to show the pooled mortality for the 37 studies (Fig 2). This confirmed the significant heterogeneity of the Egger’s test and the significant bias of the funnel plot, and identified the 22 studies falling outside the 95% pseudo-CI region of the funnel plot. Although the pooled mortality in the forest plot was 18.42% (95% CI; 14.27-22.57), there was a significant range in mortality values for Ifa 2018, 4.40% (95% CI; 1.21-7.59), to Habtamu et.al. 2021, 80.48% (95% CI; 75.12-85.84). In view of the heterogeneity, subgroup and sensitivity analyses were performed by region (Fig 5), and by age (Fig 6).

227 Subgroup analysis

228-239 Some figures in 228 to 236 do not agree with the table in 239. For example, Addis Ababa and Amhara region and SNNPR.

233 to 236 Is incorrect. The subgroup analysis is between groups not within groups. Within group heterogeneity is significant (all i^2 are high) at p=0.00, which means that the heterogeneity values are not obtained by chance. The test of group differences, that is, the between group differences shows significant heterogeneity with i^2 = 98.02. However, p=0.30 for group differences, shows that there is a 30% chance that the heterogeneity values are incorrect, which is much higher than the benchmark p=0.05 and so are not significant (once again STATA gives exact p-values in the forest plots).

239 The table should show a column between Region and Proportion ((95% CI) for study size. For example, Addis Ababa| 3 | 20.36 (1.76-36.97)| 68.22| 97.82| 0.00| and for the rest of the rows. This emphasises that some areas have very low study sizes.

246-248 Is incorrect. The test for group differences had p=0.11 which is above p=0.05 and is not significant because there is a 11% chance that the high heterogeneity values are incorrect. The within group significant heterogeneity is significant at p=0.00 or p=0.01.

256 Add that the weights in Fig 2 are approximately equal and group around 100/37=2.70, which is satisfactory and shows that no study unduly influences the overall weighting.

258 …in this study, the pooled effect size of the 20 independent predictors was assessed.

297 Incorrect. A p-value is the chance that a test is by chance or incorrect. P=0.244 means one cannot accept the significance or non-significance of the test.

380-381 our studies had the following limitations: the significant heterogeneity in the pooled proportion of mortality of the 37 studies, remained in the subgroup and sensitivity analysis and within study analysis. The funnel plot and Galbraith plot analysis indicated significant bias which was not resolved in the fill and trim procedure in the case of the funnel plot. Nevertheless the paper has significant value in establishing benchmark values in the case of a country with significant diversity in its independent predictors.

7. PLOS authors have the option to publish the peer review history of their article (what does this mean?). If published, this will include your full peer review and any attached files.

Reviewer #1: **Yes: **Anthony L H Mayne

---

## [Author Response · Author response to Decision Letter 2]

4 Nov 2024

Authors' response to the Editor, and Reviewers' concern 

Title: Mortality and predictors among HIV/TB co-infected patients in Ethiopia: A Systematic Review and Meta-analysis.

Manuscript ID: PONE-D-24-06919.

To: PLOS ONE JOURNAL

Date: November,4, 2024.

Subject: Revision of the manuscript. 

Thank you, very much dear editor and reviewers, for your valuable and constructive suggestions and comments. We found that the comments are very helpful and constructive to further improve the manuscript. This is our point-by-point response of authors to the editor and reviewers’ suggestions and concerns about the manuscript entitled “Mortality and predictors among HIV/TB co-infected patients in Ethiopia: A Systematic Review and Meta-analysis” which has a manuscript ID of “PONE-D-24-06919” given by the journal. It is known that the manuscript has been reviewed by reviewers and sent back to the authors for revision and resubmission. As authors of this manuscript, the comments and concerns raised by the reviewers and editor are highly important enabling us to improve the quality and plausibility of the manuscript. To do so, we have addressed all of the reviewer's concerns point by point. Therefore, we are very pleased to resubmit the revised version of the manuscript for further revision process and facilitation of its publication on PLOS ONE. 

We look forward to hearing from you at your earliest convenience. 

With best regards.

Corresponding Author

Wubet Tazeb Wondie E-mail: wubettazeb27@gmail.com.

On behalf of Co-authors.

Point-by-Point Response Letter

Dear editorial office of PLOS ONE we have presented our point-by-point response in a way that the reviewers' concern is depicted first, and the authors' response has been given immediately next to it.

Reviewer 1 Concerns and Authors’ Response.

Concern 1: General: reporting of data especially p-values and 95% CLs is inconsistent. STATA shows exact p-values, p=0.00 so references to forest plots must show the exact values. Sometimes you adopt the STATA representation of CIs as in the square brackets in line 246 and sometimes your depiction with round brackets. A p-value gives the probability that a statistic like i^2 is a chance value. If (say) i^2=98.002, which indicates high heterogeneity, p=0.11 is not significant, because it is higher than p=0.05 and we cannot conclude that the i^2 is significant or not (a p-value is usually computed by a chi-square distribution, and a statistic such as i2 does not determine its own p-value).

Authors response: Dear esteemed reviewer, we thank you for your valuable concern. We have accepted your suggestion and corrected it in the revised manuscript. For heterogeneity, we used Tau-squared and other statistics as shown in the forest plot in addition to the I2 value. Thanks again.

Concern 2: Lines 45-46 the lowest pooled mortality was reported for two general studies in Ethiopia 11.95 (95% CI; 4.19-19.00).

Authors' response: Dear respected reviewer, we thank you for your concern, we have accepted and corrected the revised manuscript. 

Concern 3: Line 51 … 18.42% (95% CI; 14.27-22.57), i2=98.02%. The lowest individual study mortality was 4.40 (95%CI; 1.21-7.50) and the highest, 80.48 (95%CI; 75.12-85.84), which indicated significant inter study heterogeneity. This was confirmed by the funnel and Galbraith plots, where 22 out of the 37 studies lay outside the central pseudo-95% region. The tau-squared value of 160.41 indicated there was significant within study variability. In the subgroup analysis…

Authors response: Dear respected reviewer, we thank you for your valuable concern. As you have said there was inter-study heterogeneity, and 22 out of the 37 studies lay outside the central pseudo-95% region. Accordingly, we have stated briefly this inter-study heterogeneity in the subgroup section from lines 234-258. Dear reviewer, in this regard we are ready to accept any of your recommendations. 

Concern 4: Line 65 …which accounts for 30%...

Authors response: Dear reviewer, thank you for your concern, we have accepted and corrected the revised manuscript. 

Concern 5: Line 164-167 (this is a conceptual section, not a results section). …to assess the presence of a small study effect, an Egger’s test was done. The trim and fill funnel plot analysis was confirmed by a Galbraith plot, and the studies lying inside and outside the central 95% pseudo-CI region was agreed to the distributional features in the pooled mortality forest plot in Fig 2.

Authors response: Dear respected reviewer we thank you for your important suggestions. We have accepted your suggestions and corrections are taken. 

Concern 6: Line 205-207 rewrite this (note that p values in forest plots are exact in STATA forest plots).

Authors response: Dear esteemed reviewer, thank you for your concern. We have accepted and corrected it in the revised manuscript.

Concern 7: Line 205-207 significant heterogeneity was observed between studies with a Q=123.16; and i^2 =98.02 and H^2 =50.46; P=0.00. Tau-squared=160.41 indicated significant heterogeneity within studies.

Authors response: Dear reviewer thank you for your concern. As you have mentioned, Tau squared and other statistics indicate a significant heterogeneity, and we have narrated it in the revised manuscript. 

Concern 8: Line 211-226 I suggest a rewrite of these lines as follows –

Line 211-215 An Egger’s test value, P<0.001, indicated a significant publication bias, and highly non-normal distribution. A funnel plot confirmed both the bias and the non-normal distribution, but the non-parametric trim and fill analysis failed to resolve the bias because 22 of the 37 studies and the 3 imputed studies remained outside the pseudo 95% CIs. Standard errors to effect size (mortality) varied between 6.5 standard errors and 0.5 standard errors before and after the three imputed studies. A Galbraith plot confirmed the results of the funnel plot.

Authors response: Dear esteemed reviewer, we thank you for your valuable concern. We have accepted your recommendation and corrected it in the revised manuscript.

Concern 9: Lines 216 Fig 3: Non-parametric funnel plot with fill and trim analysis

Authors response: Dear esteemed reviewer we thank you for valuable suggestions. We have accepted and corrected in the revised manuscript. 

Concern 10: Line 217 Fig 4: Galbraith plot

Authors response: Dear respected reviewer, thank you for your suggestions. We have accepted and corrected it in the revised manuscript.

Concern 11: Line 218-226 Handling heterogeneity

Because of the high heterogeneity between and within the 37 studies, a forest plot was prepared using the REML method to show the pooled mortality for the 37 studies (Fig 2). This confirmed the significant heterogeneity of the Egger’s test and the significant bias of the funnel plot, and identified the 22 studies falling outside the 95% pseudo-CI region of the funnel plot. Although the pooled mortality in the forest plot was 18.42% (95% CI; 14.27-22.57), there was a significant range in mortality values for Ifa 2018, 4.40% (95% CI; 1.21-7.59), to Habtamu et.al. 2021, 80.48% (95% CI; 75.12-85.84). In view of the heterogeneity, subgroup and sensitivity analyses were performed by region (Fig 5), and by age (Fig 6).

Authors response: Dear respected reviewer, we thank you for your valuable comments. We have accepted and corrected it in the revised manuscript. 

Concern 12: Line 227 Subgroup analysis, Line 228-239 Some figures in 228 to 236 do not agree with the table in 239. For example, Addis Ababa and Amhara region and SNNPR.

Authors response: Dear respected reviewer, thank you for your golden suggestion. We have accepted and corrected it in the revised manuscript. 

Concern 13: Line 233 to 236 Is incorrect. The subgroup analysis is between groups not within groups. Within group heterogeneity is significant (all i^2 are high) at p=0.00, which means that the heterogeneity values are not obtained by chance. The test of group differences, that is, the between group differences shows significant heterogeneity with i^2 = 98.02. However, p=0.30 for group differences, shows that there is a 30% chance that the heterogeneity values are incorrect, which is much higher than the benchmark p=0.05 and so are not significant (once again STATA gives exact p-values in the forest plots).

Authors response: Dear esteemed reviewer, we thank you for your concern. We have accepted your suggestions and corrected it in the revised manuscript. However, the I2= 98.02 is for the overall pooled estimate not for between studies, but the P-value =0.30 is for between studies heterogeneity (test of group differences). Dear reviewer in this regard, we are ready to accept any of your recommendations. 

Concern 14: Line 239 The table should show a column between Region and Proportion ((95% CI) for study size. For example, Addis Ababa| 3 | 20.36 (1.76-36.97) | 68.22| 97.82| 0.00| and for the rest of the rows. This emphasises that some areas have very low study sizes.

Authors response: Dear respected reviewer, we thank you for your valuable concern. We have accepted and corrected it in the revised manuscript.

Concern 15: Lines 246-248 Is incorrect. The test for group differences had p=0.11 which is above p=0.05 and is not significant because there is a 11% chance that the high heterogeneity values are incorrect. The within group significant heterogeneity is significant at p=0.00 or p=0.01.

Author response: Dear respected reviewer we are grateful for your golden concern, we found these concerns to be helpful. We have accepted and corrected it in the revised manuscript. 

Concern 16: 256 Add that the weights in Fig 2 are approximately equal and group around 100/37=2.70, which is satisfactory and shows that no study unduly influences the overall weighting.

Author response: Dear respected reviewer, we thank you for your insightful suggestions. We have accepted your suggestion and corrected it in the revised manuscript. 

Concern 17: 258 …in this study, the pooled effect size of the 20 independent predictors was assessed.

Authors response: Dear respected reviewer, we thank you for your suggestions. We have accepted your comment and corrected it in the revised manuscript. 

Concern 18: 297 Incorrect. A p-value is the chance that a test is by chance or incorrect. P=0.244 means one cannot accept the significance or non-significance of the test.

Author response: Dear reviewer, thank you for your valuable concern. The Eggers test result showed a P-value of 0.244 which is above the benchmark 0.05, this suggests that there is no statistically significant evidence of asymmetry in the funnel plot. This means that we don’t have strong evidence to conclude that the publication bias is present in this predictor, and we tried to show this in the manuscript. Dear reviewer, in this regard we are ready to accept any of your recommendations. 

Concern 19: 380-381 our studies had the following limitations: the significant heterogeneity in the pooled proportion of mortality of the 37 studies, remained in the subgroup and sensitivity analysis and within study analysis. The funnel plot and Galbraith plot analysis indicated significant bias which was not resolved in the fill and trim procedure in the case of the funnel plot. Nevertheless, the paper has significant value in establishing benchmark values in the case of a country with significant diversity in its independent predictors.

Authors response: Dear reviewer, we thank you for your golden concern, we found that your concern is significantly important for the clarity of our study. We have accepted your suggestions and we take corrections in the revised manuscript. 

Dear reviewer, we thank you for your valuable concerns. All of your concerns were constructive and they helped us to improve our manuscript for publication. If something is unclear/wrong, please let me know again!

We author would like to thank the Editor and the reviewer for their constructive concerns.

---

## [Decision Letter · Decision Letter 3]

22 Nov 2024

PONE-D-24-06919R3Mortality and predictors among HIV-TB co-infected patients in Ethiopia: A Systematic Review and Meta-analysis.PLOS ONE

Dear Dr. Wondie,

Thank you for submitting your manuscript to PLOS ONE. After careful consideration, we feel that it has merit but does not fully meet PLOS ONE’s publication criteria as it currently stands. Therefore, we invite you to submit a revised version of the manuscript that addresses the points raised during the review process.

Please note there are several minor changes (including some grammar issues which need correction).

We look forward to receiving your revised manuscript.

Kind regards,

Elizabeth S. Mayne, M.D.

Academic Editor

PLOS ONE

Journal Requirements:

Reviewers' comments:

Reviewer's Responses to Questions

**Comments to the Author**

1. If the authors have adequately addressed your comments raised in a previous round of review and you feel that this manuscript is now acceptable for publication, you may indicate that here to bypass the “Comments to the Author” section, enter your conflict of interest statement in the “Confidential to Editor” section, and submit your "Accept" recommendation.

Reviewer #1: (No Response)

2. Is the manuscript technically sound, and do the data support the conclusions?

Reviewer #1: Yes

3. Has the statistical analysis been performed appropriately and rigorously? 

Reviewer #1: Yes

4. Have the authors made all data underlying the findings in their manuscript fully available?

Reviewer #1: Yes

5. Is the manuscript presented in an intelligible fashion and written in standard English?

Reviewer #1: Yes

6. Review Comments to the Author

Reviewer #1: Please make these changes to items and use of capitals in the lines below

Lines

45 31.86% (95% CI: 7.69 - 56.03)

47-48 anaemia (HR = 2.25, 95% CI: 1.5 – 3.07)

49 co-trimoxazole preventive therapy (HR = 1.87, 95% CI: 1.28 – 2.73)

50 (HR = 1.23, 95% CI: 1.01 – 1.51)

66 [6]. (full stop)

107 Systematic

108 Systematic

153 predictors of mortality

236 31.86% (95% CI: 7.69 – 56.03)

237 20.73% (95% CI: 15.15 – 26.30)

237-238 20.36% (95% CI: 1.76 – 38.97)

238 15.19% (95% CI: 10.73 – 19.65) 13.66% (95% CI: 8.98 – 18.35)

239 13.74% (95% CI: 6.86 – 20.62)

240 11.95% (95% CI: 4.91 – 19.00)

242 (regions)

251 pediatrics

252 20.70% (95% CI: 14.29 – 27.11)

281 hazard ratio

290 hazard ratio

308 hazard ratio

313 co-infected

314-315 18.42% (95% CI: 14.27 – 22.57)

7. PLOS authors have the option to publish the peer review history of their article (what does this mean?). If published, this will include your full peer review and any attached files.

Reviewer #1: **Yes: **Anthony Leland Hamilton Mayne

---

## [Author Response · Author response to Decision Letter 3]

1 Dec 2024

Authors' response to the Editor, and Reviewers' concern 

Title: Mortality and predictors among HIV/TB co-infected patients in Ethiopia: A Systematic Review and Meta-analysis.

Manuscript ID: PONE-D-24-06919.

To: PLOS ONE JOURNAL

Date: December 1, 2024.

Subject: Revision of the manuscript. 

Thank you, very much dear editor and reviewer, for your valuable suggestions and comments. We found that the comments are very helpful and constructive to further improve the manuscript. This is our point-by-point response of authors to the editor and reviewers’ suggestions and concerns about the manuscript entitled “Mortality and predictors among HIV/TB co-infected patients in Ethiopia: A Systematic Review and Meta-analysis” which has a manuscript ID of “PONE-D-24-06919” given by the journal. It is known that the manuscript has been reviewed by reviewers and sent back to the authors for revision and resubmission. As authors of this manuscript, the comments and concerns raised by the reviewers and editor are highly important enabling us to improve the quality and plausibility of the manuscript. To do so, we have addressed all of the reviewer's concerns point by point. Therefore, we are very pleased to resubmit the revised version of the manuscript for further revision process and facilitation of its publication on PLOS ONE. 

We look forward to hearing from you at your earliest convenience. 

With best regards.

Corresponding Author

Wubet Tazeb Wondie E-mail: wubettazeb27@gmail.com.

On behalf of Co-authors.

Point-by-Point Response Letter

Dear editorial office of PLOS ONE we have presented our point-by-point response in a way that the reviewers' concern is depicted first, and the authors' response has been given immediately next to it.

Editor concern: Please review your reference list to ensure that it is complete and correct. If you have cited papers that have been retracted, please include the rationale for doing so in the manuscript text, or remove these references and replace them with relevant current references. Any changes to the reference list should be mentioned in the rebuttal letter that accompanies your revised manuscript. If you need to cite a retracted article, indicate the article’s retracted status in the References list and also include a citation and full reference for the retraction notice.

Author response: Dear respected Editor, we thank you for your golden concern. Based on your recommendation, we have reviewed all of the references, and none of them have been retracted. All references from journals or web pages are available. Thanks again. 

Reviewer 1 Concerns and Authors’ Response.

Concern 1: Line 45, 31.86% (95% CI: 7.69 - 56.03)

Authors response: Dear esteemed reviewer, we thank you for your valuable suggestions. We have accepted and corrected the revised manuscript.

Concern 2: Line,47-48. anaemia (HR = 2.25, 95% CI: 1.5 – 3.07)

Authors response: Dear respected reviewer, thank you for your concern, we have accepted your suggestion and corrected it in the revised manuscript.

Concern 3: Line 49, co-trimoxazole preventive therapy (HR = 1.87, 95% CI: 1.28 – 2.73). 

Authors response: Thank you dear reviewer for your comments. We have accepted and corrected it in the revised manuscript.

Concern 4: Line 50 (HR = 1.23, 95% CI: 1.01 – 1.51)

Authors response: Thank you dear reviewer for your suggestions. We have accepted your suggestions and corrected the revised manuscript.

Concern 5: Line 66 [6]. (full stop)

Authors response: Dear respected reviewer, thank you for your valuable suggestions. We have accepted your suggestion and corrected it in the revised manuscript. 

Concern 6: Line 107 Systematic

Authors' response: Dear reviewer, we thank you for your suggestion, we have accepted and corrected it. 

Concern 7: Line 108 Systematic

Authors response: Thank you, dear reviewer, for your suggestions we have accepted and corrected it.

Concern 8: Line 153 predictors of mortality

Authors response: Dear esteemed reviewer, thank you for your comment, we have accepted and corrected in the revised manuscript.

Concern 9: Line 236 31.86% (95% CI: 7.69 – 56.03)

Authors response: Dear esteemed reviewer, thank you for your comment, we have accepted and corrected the revised manuscript.

Concern 10: Line 237 20.73% (95% CI: 15.15 – 26.30)

Authors response: Authors response: Dear esteemed reviewer, thank you for your comment, we have accepted and corrected the revised manuscript.

Concern 11: Line 237-238 20.36% (95% CI: 1.76 – 38.97).

Authors response: Dear esteemed reviewer, thank you for your comment, we have accepted and corrected the revised manuscript.

Concern 12: Line 238 15.19% (95% CI: 10.73 – 19.65) 13.66% (95% CI: 8.98 – 18.35)

Authors response: Dear respected reviewer, thank you for your concern, we have accepted and corrected the revised manuscript.

Concern 13: Line 239 13.74% (95% CI: 6.86 – 20.62)

Authors response: Thank you dear esteemed reviewer for your suggestions, we have accepted and corrected it.

Concern 14: Line 240 11.95% (95% CI: 4.91 – 19.00)

Authors response: Dear esteemed reviewer, thank you for your comment, we have accepted and corrected it in the revised manuscript.

Concern 15: Line 242 (regions)

Authors response: Dear reviewer, we thank you for your valuable suggestions, we have accepted and corrected it.

Concern 16: Line 251 pediatrics

Authors response: Thank you dear esteemed reviewer for your suggestions, we have accepted and corrected the revised manuscript.

Authors' response: Dear respected reviewer, we thank you for your concern, we have accepted and corrected the revised manuscript.

Concern 17: Line 252 20.70% (95% CI: 14.29 – 27.11)

Authors response: Dear reviewer, we thank you for your valuable suggestions, we have accepted and corrected it.

Concern 18: Line 281 hazard ratio

Authors response: Dear esteemed reviewer, thank you for your comment, we have accepted and corrected the revised manuscript.

Concern 19: Line 290 hazard ratio

Authors' response: Dear reviewer, we thank you for your valuable suggestions, we have accepted and corrected it in the revised manuscript.

Concern 20: Line 308 hazard ratio

Authors response: Dear respected reviewer, thank you for your valuable comment, we have accepted and corrected it.

Concern 21: Line 313 co-infected

Authors' response: Dear reviewer, we thank you for your suggestions, we have accepted and corrected it.

Concern 22: Line 314-315 18.42% (95% CI: 14.27 – 22.57) 

Authors response: Dear esteemed reviewer, thank you for your valuable suggestion, we have accepted and corrected it.

Dear reviewer, we thank you for your valuable concerns and time. All of your concerns were constructive and they helped us to improve our manuscript for publication. If something is unclear/wrong, please let me know again!

We author would like to thank the Editor and the reviewer for their constructive concerns.

---

## [Decision Letter · Decision Letter 4]

20 Dec 2024

Mortality and predictors among HIV-TB co-infected patients in Ethiopia: A Systematic Review and Meta-analysis.

PONE-D-24-06919R4

Dear Dr. Wondie,

We’re pleased to inform you that your manuscript has been judged scientifically suitable for publication and will be formally accepted for publication once it meets all outstanding technical requirements.

Kind regards,

Elizabeth S. Mayne, M.D.

Academic Editor

PLOS ONE

Additional Editor Comments (optional):

Reviewers' comments:

Reviewer's Responses to Questions

**Comments to the Author**

1. If the authors have adequately addressed your comments raised in a previous round of review and you feel that this manuscript is now acceptable for publication, you may indicate that here to bypass the “Comments to the Author” section, enter your conflict of interest statement in the “Confidential to Editor” section, and submit your "Accept" recommendation.

Reviewer #1: All comments have been addressed

2. Is the manuscript technically sound, and do the data support the conclusions?

Reviewer #1: Yes

3. Has the statistical analysis been performed appropriately and rigorously? 

Reviewer #1: Yes

4. Have the authors made all data underlying the findings in their manuscript fully available?

Reviewer #1: Yes

5. Is the manuscript presented in an intelligible fashion and written in standard English?

Reviewer #1: Yes

6. Review Comments to the Author

Reviewer #1: (No Response)

7. PLOS authors have the option to publish the peer review history of their article (what does this mean?). If published, this will include your full peer review and any attached files.

Reviewer #1: **Yes: **Anthony Leland Hamilton Mayne

---

## [Editor Report · Acceptance letter]

26 Dec 2024

PONE-D-24-06919R4 

PLOS ONE

Dear Dr. Wondie, 

I'm pleased to inform you that your manuscript has been deemed suitable for publication in PLOS ONE. Congratulations! Your manuscript is now being handed over to our production team.

Kind regards, 

on behalf of

Dr. Elizabeth S. Mayne 

Academic Editor

PLOS ONE